# Satellite-based emergency mapping using optical imagery: Experience and reflections from the 2015 Nepal earthquakes

Jack G. Williams[1], Nick J. Rosser[1], Mark E. Kincey[1], Jessica Benjamin[1], Katie J. Oven[1], Alexander L. Densmore[1], David G. Milledge[1], Tom R. Robinson [1], Colm A. Jordan[2], and Tom A. Dijkstra[2,3]

[1] Institute of Hazard, Risk and Resilience and Department of Geography, Durham University, Lower Mountjoy, South Road, Durham. DH1 3LE. United Kingdom.
[2] British Geological Survey, Natural Environment Research Council, Environmental Science Centre, Keyworth, Nottingham. NG12 5GG. United Kingdom.
[3] School of Architecture, Building and Civil Engineering, Loughborough University, Ashby Road, Loughborough. LE11 3TU. United Kingdom.

*Correspondence to*: Jack G. Williams (j.g.williams@durham.ac.uk)

**Abstract.** Landslides triggered by large earthquakes in mountainous regions contribute significantly to overall earthquake losses and pose a major secondary hazard that can persist for months or years. While scientific investigations of coseismic landsliding are increasingly common, there is no protocol for rapid (hours-to-days) humanitarian-facing landslide assessment, and no published recognition of what is possible and what is useful to compile immediately after the event. Drawing on the 2015 $M_w$ 7.8 Gorkha earthquake in Nepal, we consider how quickly a landslide assessment based upon manual satellite-based emergency mapping (SEM) can be realistically achieved, and review the decisions taken by analysts to ascertain the timeliness and type of useful information that can be generated. We find that, at present, many forms of landslide assessment are too slow to generate relative to the speed of a humanitarian response, despite increasingly rapid access to high-quality imagery. Importantly, the value of information on landslides evolves rapidly as a disaster response develops, so identifying the purpose, timescales, and end-users of a post-earthquake landslide assessment is essential to inform the approach taken. It is clear that discussions are needed on the form and timing of landslide assessments, and how best to present and share this information, before rather than after an earthquake strikes. In this paper, we share the lessons learned from the Gorkha earthquake, with the aim of informing the approach taken by scientists to understand the evolving landslide hazard in future events and the expectations of the humanitarian community involved in disaster response.

**Keywords:** Coseismic landslides, Satellite-based emergency mapping, Landslide mapping, Disaster response

## 1 Introduction

### 1.1 Mapping landslides after earthquakes

Landsliding is a significant secondary earthquake hazard that can account for up to 25% of earthquake fatalities in mountainous regions (Yin et al., 2009; Budimir et al., 2014). In addition, the collateral damage and disruption caused by landslides

substantially inhibits short- and medium-term relief efforts by blocking or destroying transport corridors and communications (Bird and Bommer, 2004; Pellicani et al., 2014; Robinson et al., 2015). The assessment of landslide extent and impacts, beyond direct observations on the ground (Collins and Jibson, 2015; Tiwari et al., 2017), relies on the following three approaches: (1) empirical modelling, which uses a combination of pre-earthquake topographic data and information on ground motion and

shaking intensity, (2) manual landslide mapping, and (3) automated landslide mapping, both of which use post-earthquake airborne or satellite remote sensing. The main outputs from these assessments are maps of landslide locations, extents and densities, the humanitarian value of which is widely recognized (e.g. Goodchild, 2007).

Each approach has specific data requirements, with the capture and appraisal of those data resulting in an inevitable latency between the event and the release of information (UN-SPIDER, 2015; Fleischhauer et al., 2017). For manual mapping, the

speed of information production is influenced by the nature of the landslides themselves, the data quality, and choices about what and how to map (Joyce et al., 2009). Although critical for defining the speed of the assessment, those choices have not previously been described or evaluated with respect to the timescales of the information needs of those on the ground. However, the potential value is clear: if available within a very short timeframe (hours-to-days), information on landsliding can be highly beneficial.

Recently, considerable gains have been made in the capture of satellite imagery used for landslide assessment, particularly in terms of: (1) the resolution and bandwidth of the sensors used; (2) the spatial and temporal coverage; and (3) the ease of access via online repositories (Voigt et al., 2016). However, no single automated method exists to map landslides in rapid response assessments due to the complexities and variability between earthquakes in different locations (Casagli et al., 2016), resulting in uncertainty regarding the type and timeliness of information that is useful to produce. Standards or guidelines for Satellite-

based Emergency Mapping (SEM) have been developed for some hazards, such as flooding (UN-SPIDER, 2015; Voigt et al., 2016), and mechanisms such as the EU Copernicus Management Service have provided specifications for the creation of rapid mapping products after disasters, including landslides. Despite these advances, clear and widely accepted guidelines for humanitarian-facing landslide assessments have not yet been developed, yet are essential for defining open, constructive and ethical approaches to SEM.

While many satellite operators have tasked rapid image capture of earthquake-affected areas, either on humanitarian grounds via established international frameworks (e.g. the International Charter on Space and Major Disasters) or for commercial ends (Joyce et al., 2009), the use of these data is not necessarily coordinated. For example, commercial satellite imagery at sub-meter resolution was released for the benefit of the response to the 2010 Haiti earthquake (Harp et al., 2011). Over 300 map products were created within two weeks by a plethora of agencies, each using different procedures and standards (UN-SPIDER,

2015; Voigt et al., 2016). Uncoordinated mapping efforts undertaken with different objectives, and for different end-users, can result in a duplication of effort and may cause confusion and data saturation amongst the humanitarian response community. This has the potential to produce an incomplete and inconsistent assessment of humanitarian need (IASC, 2012). In the longer term, these initiatives can result in multiple inventories for the same event, further adding to the confusion. For example, Xu (2015) described eight separate landslide inventories compiled after the 2008 Wenchuan earthquake in China. After the 2015

Nepal earthquakes, there was a five-fold increase in landslide numbers between the inventories reported by Kargel et al. (2016; 4 312), Martha et al. (2016; 15 551), Roback et al. (2017; 24 915), and Tiwari et al. (2017; 14 670). While some of these inventories were created in the immediate aftermath of the disaster, their use for scientific purposes nevertheless assumes complete coverage of the affected area. The resolution of mapping and the approach taken should therefore be stated clearly

alongside the purpose of the inventory.

## 1.2 The need for rapid landslide assessment

Previous research has defined appropriate scientific methods for coseismic landslide mapping (e.g. Gorum et al., 2011; Harp et al., 2011; Wasowski et al., 2011; Guzzetti et al., 2012), with some organizations, such as UNITAR/UNOSAT and EU Copernicus requesting feedback from end-users. However, there remains an absence of readily available information on what

is actually useful for decision makers who are tasked with dealing with an earthquake and its cascading hazards, particularly where rapid response times are key. Underpinning the effort we describe below is the broad timeframe of a humanitarian disaster response, based upon United Nations disaster response protocols. Central to this is the 'Humanitarian Needs Assessment', which aims to *'provide fundamental information on the needs of affected populations and to support the identification of strategic humanitarian priorities'* (IASC, 2012:4). This approach to disaster response starts immediately after

an earthquake and comprises a *Situation Analysis* (completed within 72 hours) and a *Multi-Sector/Cluster Initial Rapid Assessment (MIRA) Report* (completed within two weeks; IASC, 2015). During the first phase, emphasis is placed on obtaining pre- and post-crisis data to determine the disaster extent and scale. This phase '*balances the need for accuracy and detail with the need for speed and timeliness'* (OCHA, 2013) and informs the basis of the mapping approach described below. The UN approach emphasizes the need for information that is fit for purpose, such that superfluous detail and precision are actively

discouraged (OCHA, 2013) .

While coseismic landslide inventories created for academic research are slowly and painstakingly collected, this approach is likely to be inconsistent with the requirements for rapid, widespread coverage and the identification of broad areas of concern. The need is therefore to identify the areal extent and location of landsliding (*scale* and *intensity*), assess how landsliding intersects with the location of people and infrastructure (*impacts*), and to appraise the residual risks from induced hazards

(*priorities*), such as existing or potential landslide dams. These needs must be balanced against the type and timeliness of information that can be produced. Post-earthquake end-users of landslide information can be numerous, with complex responsibilities, requirements, and information needs. These requirements are also highly dynamic, often shifting from a broad-scale impact assessment to increasingly local level detail over a matter of days, and are therefore challenging to satisfy through SEM (Voigt et al., 2016). As a consequence, the utility of particular forms of information evolves from the initial response to

the early recovery. Importantly, the time necessary to produce some forms of information may render them redundant in the context of the initial response, and therefore unnecessary to produce rapidly.

Here we examine these general issues by focussing on the case of the 2015 Gorkha earthquake and its aftershocks, which triggered thousands of landslides in Nepal. Given the steep terrain, the large rural population, and reported initial shaking

intensities in Nepal, the potential for landslide-induced losses as a result of the 2015 earthquakes was quickly recognized (e.g. Gallen et al., 2016; Robinson et al., 2017). We reflect upon a rapid landslide assessment that was undertaken over the first two months after the earthquake, and efforts to disseminate the findings to potential end-users in Nepal and elsewhere. We consider the benefits and time needed for various assessments of landsliding that range from rapid appraisal to a full inventory, enabling

an evaluation of the approaches that can effectively inform critical decision-making. We also consider the methods that we applied to expedite the generation of usable outputs, which were often at odds with the practices associated with collating a formal scientific landslide inventory. We close by offering recommendations for conducting future humanitarian need-driven rapid landslide assessments following a large earthquake.

## 2 Methods

### 2.1 Initial landslide identification efforts

Our mapping efforts were undertaken by a group of five analysts from Durham University and three from the British Geological Survey (BGS), with experience of conducting landslide research in Nepal or similar terrains. The assessments fed information to, and were guided by, the needs of humanitarian actors in Nepal, including the UN Resident Coordinator's Office in Kathmandu and members of the Nepal Risk Reduction Consortium (NRRC), as well as the Cabinet Office Briefing Room

(COBR), the Scientific Advisory Group for Emergencies (SAGE), the Foreign and Commonwealth Office (FCO), and DFID (Department for International Development) in the UK. Contacts in Nepal were well established because of a long-term collaborative project, Earthquakes without Frontiers (see: ewf.nerc.ac.uk), which brought together natural and social scientists, policy makers, and practitioners with the aim of building societal resilience to earthquakes and associated secondary hazards. Contacts with UK Government departments were also well established because of prior provision of advice for a range of

global hazards. These contacts enabled a more rapid assessment of the type of information required during the response. Decisions on how to assess the coseismic landslides invariably related to how and where to map landsliding, and what to map. Based on the need to inform the humanitarian response, and directed by requests from the UK Government, our assessments focused on the relatively populous middle Himalaya of western and central Nepal, where any landslides were more likely to directly affect people and infrastructure. We also mapped portions of the High Himalaya because of the potential for substantial

downstream impacts, such as flooding from breached landslide dams. Initial searches for landslide dams were therefore paramount, and dams that were identified were monitored until breached. This effort ran in parallel to several other initiatives that have subsequently been reported (Kargel et al., 2016; Roback et al., 2017; Tiwari et al., 2017).

The mainshock, which generated the majority of landslides (Martha et al., 2016; Roback et al., 2017), occurred on the Main Himalayan Thrust (MHT) with $M_w$ 7.8 and an epicenter in Gorkha District in Western Nepal. The rupture propagated

eastwards, impacting areas up to ~140 km from the epicenter, with additional large aftershocks concentrated near the eastern end of the mainshock rupture plane (Avouac et al., 2015; Galetzka et al., 2015). A rapid appraisal of the first available imagery suggested that landsliding occurred in an E-W swath located north of the Kathmandu Valley, covering a large proportion of

Western and Central Nepal (~12 000 km$^2$). Initial indications from coseismic earthquake-triggered landslide models, based on Kritikos et al. (2015) and Parker et al. (2017), were used to direct the mapping effort (see: http://ewf.nerc.ac.uk/2015/04/25/nepal-earthquake-likely-areas-of-landsliding/). However, mapping efforts were constrained by widespread cloud cover that limited the availability of good-quality optical imagery.

**2.2 Optical image selection**

Landslides are most identifiable in optical satellite images under daytime conditions with minimal shadow and cloud, captured at a time of year when vegetation and landslides produce a sharp radiometric contrast. From experience, such conditions are rarely coincident or likely. Given that landslides typically occur in steep and mountainous regions, often following prolonged rainfall, the potential for cloud cover in imagery is a key consideration for associated SEM. The Nepal Himalaya, for example,

are obscured by cloud between mid-June and mid-September each year, during which time an estimated 90% of annual fatal landsliding occurs (Petley et al., 2007). Landslide inventories conventionally draw on a full catalogue of imagery compiled before mapping begins to ensure consistent coverage of the entire area (Harp et al., 2011). Ideally, all images are collected by a single sensor, providing consistent spatial, spectral and radiometric resolution appropriate for the type of landsliding under investigation. A key challenge of time-critical SEM responses is the selection of the most effective imagery for mapping. This

selection must be made before complete knowledge of post-earthquake imagery can be acquired, and usually before the general spatial distribution of landsliding is known. Most commonly, imagery from a variety of sensors is captured iteratively, and is distributed across multiple on- and off-line repositories and platforms. Efficient mapping from this data requires a method for selecting the most 'useful' images, which demands that attributes such as the minimum swath width, maximum topographic distortion, and desired spatial, spectral and radiometric resolutions are defined. The nature of the terrain, the ground cover, and

the style of landsliding therefore hold considerable influence over the necessary requirements of imagery that is useful for mapping.

Consequently, as part of our effort, a protocol for prioritizing imagery from which to map was developed (Fig. 1). It quickly became apparent that, given the number and spatial extent of landslides and the need for mapping consistency, beginning to map from a new image committed one mapper for a considerable amount of time. During this time, it was increasingly probable

that better imagery of the same area would become available. Imagery was therefore prioritized by three criteria: (1) the platform and hence speed with which the imagery could be handled and analysed; (2) characteristics of the imagery, including cloud cover and geometric distortion; and (3) the spatial and spectral resolution, as well as the swath width. These criteria were used to develop a decision-tree structure for efficient image selection that is described in Fig. 1.

**2.2.1 Mapping platform**

Efficient mapping requires a platform for quick navigation and mapping of large quantities of images, or a way of bypassing the need for georeferencing. The image source, and hence the platform, influenced which images were prioritized due to the

relative ease with which mapping could be conducted as compared to downloading, pre-processing, and mapping from raw imagery. While this made the mapping more fragmented, the mapping time was substantially reduced.

Two platforms were employed for image interpretation: ESRI's ArcMap and Google Earth™. ERDAS Imagine and ENVI were used in the BGS to process the raw satellite images and convert them to full resolution lossless compressed formats prior
to making them available for interpretation in ArcMap. Mapping within ArcMap was somewhat problematic for several reasons. WorldView-2 and WorldView-3 GeoTIFFs are large files (~1.4 GB panchromatic, ~0.8 GB multi-spectral), and therefore required considerable time for pyramid construction and were hampered by stilted image refresh rates, each of which hindered the speed of mapping. Medium-resolution downsampled JPEGs (~100 MB) were therefore downloaded from the USGS HDDS Explorer as an alternative to increase mapping speed. This reduction in file size equated to a decrease in cell
size from ~0.3-0.5 m to ~3-4 m, preserving the ability to map most failures. Due to the lack of orthorectification, however, geolocation errors in the JPEG imagery were up to 3 km.

To reduce georeferencing times, we used the DigitalGlobe™ online platform to view WorldView imagery, running alongside Google Earth™ to view imagery provided by Google Crisis Response, which included DigitalGlobe™ WorldView-2, WorldView-3, and Airbus Pléiades imagery. DigitalGlobe's platform provided the timeliest access to orthorectified
WorldView imagery, enabling a rapid assessment of the degree of cloud cover and the extent of landsliding in those areas that had previously been obscured but without the capacity to map onto the images. Access to Google Crisis imagery provided additional benefits: (1) pre-earthquake imagery was readily available to distinguish new and reactivated landslides; (2) image navigation and zooming were quicker than in ArcMap; (3) the capacity for 3D panning and tilting allowed easier identification of landslides; and (4) despite the introduction of geolocation errors (Sato and Harp, 2009), landslides could be digitized and
exported into other software.

The use of both ArcMap and Google Earth™ enabled efficient handling of a large array of images of varying extent, resolution and cloud-cover. Google Crisis imagery in Google Earth™ also allowed rapid comparison of multispectral and panchromatic data to identify landslides and better delineate their extent. Despite the relative benefits of Google Crisis, it is important to note that both the georeferencing and orthorectification of imagery were poor owing to image incidence angle and cloud cover.
Poor georeferencing made it almost impossible to map by switching between multiple images for a given area of interest, which would otherwise have been a fast and effective mapping strategy. Furthermore, Google Crisis was insufficient as a standalone tool due to geolocation errors and the slow imagery update rate compared to HDDS Explorer. The primary benefit of Google Crisis was the relative ease and speed of operator use, which increased mapping speed once suitable images were available.

**2.2.2 Image and sensor characteristics**

The second criterion related to the quality of imagery, and was determined primarily by the degree of cloud cover as well as the sensor incidence angle off-nadir. Imagery with minimal cloud cover was prioritized in order to observe as much of the ground as possible within a short period of time and to minimize the time spent on georeferencing. None of the post-earthquake

images were completely cloud-free and so mapping was undertaken from multiple images wherever practicable in order to develop a mosaic of coverage. It was especially imperative to distinguish between unmapped areas obscured by cloud cover from mapped areas with no landslides. The angle off-nadir was considered because georeferencing time increased (and accuracy decreased) with increasing angle. Critically for earthquake-triggered landslides, initial data acquisition is commonly

focused at the published epicenter, rather than across the full extent of ground shaking. During the initial phases of the response, satellites were tasked to capture images centred on the epicentral region that lay south and west of the most intensive areas of landsliding further to the north. Images to the north and east were therefore captured with relatively high incidence angle off-nadir. This resulted in significant topographic occlusion and image distortion, exacerbated by the steep topography (Roback et al., 2017).

Given the prevalence of cloud cover and off-nadir viewing angles, imagery was drawn upon from a wide range of sensors, including Cartosat, DMCii, EO-1, GeoEye, Landsat, Pléiades, RapidEye, SPOT, and WorldView. Based upon the mountainous areas of Nepal that experienced moderate to severe shaking, as estimated by ShakeMap, the area of shaking sufficient to trigger landslides was approximated at 35 000 km$^2$. This estimate was supplemented by the spatial distribution of modelled landslide probabilities > 0.5 (see: http://ewf.nerc.ac.uk/2015/04/25/nepal-earthquake-likely-areas-of-landsliding/). With the exception

of the EO-1 Advanced Land Imager (ALI) and Landsat 8, the underline{swath width} of sensors such as WorldView-2 (16.4 km at nadir) and WorldView-3 (13.1 km at nadir) was small in comparison to this area, and so large numbers of relatively small-footprint images were needed for complete coverage. Where possible, images with large areal extents were therefore selected to gain a synoptic overview. The time taken to georeference several hundred images, and the varying degrees of success (RMSE of up to ~60-140 m in most areas except for the valley floors), made it unfeasible to process and map imagery fast enough to keep

pace with its release. While having a high spatial resolution (~3 m) and short return period, PlanetLabs imagery had a small image footprint (~50 km$^2$) relative to the affected area. The low radiometric performance of this imagery (Houbourg and McCabe, 2016) also hindered landslide identification in comparison to sensors, such as EO1-ALI.

underline{Spectral resolution} and contrast were also used in selecting suitable images. Given our observation that most landslides were shallow and comprised rockfalls and shallow rockslides, spectral resolution and, in particular, the presence of a NIR band were

of considerable importance in landslide mapping. These were prioritized over spatial resolution as long as the latter remained commensurate with the size of landslides. In the case of WorldView-2 and WorldView-3, although panchromatic imagery provides greater spatial resolution, the ability to distinguish vegetation from freshly exposed bedrock and regolith in landslide scars was reduced due to the lack of multispectral imagery.

The final criterion was the underline{spatial resolution} of imagery. Most large (> 100 m length or width) landslides were observable

using the coarsest spatial resolution imagery available (Landsat 8; 30 m visible and NIR but routinely pan-sharpened to 15 m). In catchments with high drainage density, smaller landslides have the potential to block steep, narrow valleys and therefore required very high resolution (VHR; < 2 m) imagery to be delineated. For detailed mapping at a level where the proximity of landslides to infrastructure is important, VHR imagery is also needed. Medium-resolution imagery, however, still proved useful for two reasons. First, Landsat 8 imagery acquired on 2 May (one week after the mainshock) coincided with widespread

cloud-free conditions, providing the first spatially consistent synoptic dataset across the entire affected area. Second, consistency in the geolocation of multispectral data could be maintained by applying transformations used in georeferencing higher-resolution panchromatic data, in which the identification of ground control points (GCPs) between pre- and post-earthquake imagery was more accurate.

## 2.3 Mapping protocol

For consistency, most landslide inventories adopt a single method of landslide delineation (i.e., as points, polylines, or polygons), depending upon the type of output and the scale of the event. It is also common to identify individual landslides, rather than delineate areas impacted by multiple landslides (Guzzetti et al., 2012; Marc and Hovius, 2015). In global landslide databases (e.g. Kirschbaum et al., 2010; Petley, 2012) and many coseismic landslide inventories, landslides are specified as point features as an efficient means to locate and count large numbers of landslides (Kargel et al., 2016; Tiwari et al., 2017). Regional- to local-scale landslide inventory maps tend to document landslides as polygons, which can be used to understand impact zones or to separate source from deposit (Guzzetti, 2004; Guzzetti et al., 2012). Polygons are required where assessments of landslide area and volume, sediment yield, or connectivity of landslide deposits to the fluvial network are needed (e.g. Roback et al., 2017). The focus at the BGS was on mapping polygons, while the initial focus of the Durham effort was the collection of point data, which was subsequently expanded to polylines. The decision to collect point data at Durham was based on the need for rapid analysis and the large numbers ($10^3$ to $10^4$) of landslides, anticipated from previous earthquakes of similar magnitudes, such as the 2008 Wenchuan (China) earthquake that generated ~200 000 landslides (e.g. Xu, 2015). The subsequent decision to construct polylines reflected our observation that most of the landslides comprised rockfalls, shallow rockslides, and dry debris flows and avalanches, which often followed pre-existing channels and had highly elongated footprints. The time cost associated with mapping polylines, rather than points, was found to be small relative to the step from points to polygons, while the elongated landslide footprints yielded considerable information on landslide sizes and runout. Our minimum landslide size generally had a major axis of > 50 m. The method evolved iteratively as data became available and the scale and nature of the landsliding became apparent, the chronology of which is described below.

## 3 Results

### 3.1 Chronology of rapid landslide assessment using optical imagery

The chronology of selected image release, cloud cover, mapping, and released reports is provided in Fig. 2. Within 48 hours of the 25 April mainshock, initial estimates of the likely geographical distribution of landslides were based upon the outputs of the USGS ShakeMap and a limited number of reports from the ground (e.g. via social media). Although this provided a first-order approximation of potential landslide locations, coseismic landsliding is determined by the interactions between topography, ground shaking, and local site geology (Meunier et al., 2008; Parker et al., 2015; Marc et al., 2016). Empirical landslide susceptibility models (Gallen et al., 2016; Parker et al., 2017; Robinson et al., 2017) provided probabilistic estimates

of the likelihood of a landslide at any point in space within the affected area. These models predicted that landslide probabilities were high but also variable across the affected districts, especially in the middle to high Himalaya north and east of the epicenter where topographic relief increases, but where population densities remain high. Estimates provided by the USGS ShakeMap, upon which such models rely, underwent several refinements within the first 48 hours, resulting in minor alterations to model predictions, but the overall spatial distribution of relative landslide density remained unchanged. Comparisons between predicted landslide density and observed landslide density have since highlighted some important discrepancies (Gallen et al., 2016), including an overestimation of landsliding to the south of Kathmandu in the Sivalik Hills

## 3.2 27 April – 2 May: Direct landslide mapping

Prior to 2 May, cloud cover limited the availability of useable imagery across the entire affected area. During this period, two approaches were undertaken to locate landslides and to prioritize areas for mapping once cloud-free imagery became available. Estimates of landslide location and qualitative size (small/medium/large) were collated from photographs and footage posted on social media and, later, from airborne video from the news media. Although only ~20 landslides were identified and located in this manner, most were in areas north of Kathmandu and at some distance from the epicenter. Secondly, small gaps in cloud cover provided useful indicators of the extent and intensity of landsliding. For example, a small gap in cloud cover of ~1 km$^2$ in a tributary of the Upper Bhote Kosi Valley in Sindhupalchok District allowed a particularly high number of landslides to be identified in this small area (~25 km$^2$). This gap in cloud was ~120 km from the epicenter and provided an initial assessment of the nature, type and density of landsliding in the area, as well as supporting modelled estimates of the area affected by landsliding.

## 3.3 After 2 May: Landslide assessment using optical imagery

From 2 May onwards, more frequent small breaks in cloud cover provided useful image coverage in a limited but increasing number of locations. Cloud cover was often concentrated around high elevation topography, leaving valley bottoms visible. Mapping of individual landslides therefore focused in areas proximal to the channel network and lower elevation slopes to survey for landslide dams, similar to those triggered by the 2008 Wenchuan earthquake (Cui et al., 2009; Xu et al., 2014).

In order to rapidly map as large an area as possible, and due to cloud cover on higher ground, each landslide was initially marked as a single point at the toe, where the risk to infrastructure and likelihood of valley blocking was greatest. The imagery that was available during this phase had generally high off-nadir viewing angles and so geolocation errors after orthorectification were lower close to valley bottoms. In instances where the landslide toe ran out to but did not block the channel network, a 'yes/no' attribute was added describing the potential for the deposit to block the valley. In instances where upstream pooling of water and a restricted flow downstream was identified indicating blockage, a separate valley-blocking marker was created (Fig. 3). These locations were fed to the USGS for visual inspection as part of their assessment of present and future landslide hazards (Collins and Jibson, 2015).

Valleys with particularly intense landsliding were recorded with a polyline running up river from the southernmost visible extent of landsliding (Fig. 3). The aim of this was to delineate the southernmost limit of major landslide disruption, and hence the likely northern limit of unimpeded road access, using the predominantly north – south oriented drainage network. This was mapped as a solid line where the limit was observed and a dashed line where the limit was inferred in the absence of imagery.

Subsequent mapping showed this line to be an accurate estimate, with the area of intense landsliding (~12 000 km$^2$) matching our own final product and that of Roback et al., 2017 (Fig. S1). A map containing this information was released on 4 May, approximately two days after cloud cover reduced and nine days after the mainshock (Fig. 3).

As increasingly cloud-free imagery became available, manual mapping speeds increased. Landslides were subsequently identified with polylines to provide an attribute of scale and to define where landslides intersected infrastructure, such as roads.

A record of areas mapped and areas obscured by cloud was maintained. Mapping using VHR imagery identified that the majority of coseismic landslides were narrow (~10 m) and hence would be difficult to identify in lower resolution imagery. Updated maps were published online on 7 May (Fig. 4) and 21 May (Fig. 5), which featured both increasing numbers and coverage of landslides.

Our accompanying notes (an example of which is provided in Table 1) summarised the key observations, the methods used,

and key messages about the intensity, locations and general risks posed by these landslides. The maps and underpinning data were disseminated as Google Earth$^{TM}$ KML files and ArcGIS shapefiles on the Humanitarian Data Exchange Nepal (https://data.humdata.org/group/nepal-earthquake). In addition, *.PDF versions of district-level landslide maps in colour and black and white, alongside interpretive notes in English and Nepali, were posted on the Earthquakes without Frontiers blog (http://ewf.nerc.ac.uk/2015/05/28/nepal-updated-28-may-landslide-inventory-following-25-april-nepal-earthquake/) and the

National Society for Earthquake Technology website (http://www.nset.org.np/eq2015/), as well as being sent directly to the UN RCO and Nepal Red Cross. A range of PDF maps, shapefiles and reports were also posted on the BGS website (http://www.bgs.ac.uk/research/earthHazards/epom/Nepalearthquakeresponse.html) as well as sites of international organisations that provided data, such as UNOSAT (https://unosatgis.cern.ch/live/EQ20150425NPL/) and the Disasters Charter (see: https://www.disasterscharter.org/). This information was later used in, for example, UN-led monsoon

preparedness planning, and by the military in their assessment of road access constraints (Datta et al., forthcoming).

Approximately 5 600 coseismic landslides were identified in the affected area by 18 June, 42 days after the earthquake. This comprised ~4 500 triggered by the 25 April Gorkha earthquake, ~300 by the 12 May Dolakha earthquake, and ~800 that could be attributed to either event. Some areas remained obscured by clouds throughout this period and were therefore recorded as such in our final map (Fig. 6).

# 4 Discussion

## 4.1 Comparison of landslide mapping

Comparing our rapidly-derived inventory with subsequent, independently collated inventories (Martha et al., 2016; Roback et al., 2017; Tiwari et al., 2017) shows that our inventory underestimated the total number of landslides by up to ~19 000. When compared for every 1 km$^2$ of landslide-affected area (as identified in both inventories), our inventory underestimates landslide number by an average factor of 1.8, which is broadly consistent irrespective of landslide density. However, the spatial pattern and relative intensity closely adheres to those described in both Martha et al. (2016) and Roback et al. (2017). The overall extents of the mapped landslide affected area are broadly similar (Fig. S1), covering the same geographical footprint. In addition, the locations of highest density landsliding and the southernmost limit of landsliding are consistent between the inventories. The inventory therefore holds value as a rapid assessment of the relative intensity of landsliding and its spatial distribution, and as a tool for identifying the worst affected areas. This raises questions about the value of time invested in rapidly assessing metrics that are considered useful for informing disaster response, such as absolute landslide numbers and volumes, except in cases where information has been requested for specific locations. Below, we discuss the utility of such metrics in terms of the benefit of the extra detail they provide compared to the increased time required to derive them. This is an attempt to identify and develop common standards for rapid SEM for landslide-triggering events that can effectively inform the humanitarian response phase of the disaster lifecycle. Prior to this, it is important to consider the wider application of the SEM approach described above.

The approach was heavily determined by the scale of the rupture and the presence of cloud cover in the run up to the South Asian monsoon, both of which necessitated the collection of a considerable number of images and a means of prioritising them. In drier regions, or following earthquakes or rainfall that affect a much smaller area, the chronological order of outputs is unlikely to change. However, the offset in timing between initial landslide models and the mapping of landslides using either radar or optical satellite imagery is likely to decrease. The 2016 Kaikoura earthquake, New Zealand, ruptured an area 200 × 60 km in size, similar to the 120 × 80 km rupture during the Gorkha earthquake. Due to cloud-free conditions and the availability of short return interval Sentinel-2 imagery, a preliminary landslide map of 1 092 landslides was released three and a half days after the earthquake with a subsequent map of 5 875 landslides within two weeks (Sortiris et al., 2016). A smaller affected area and absence of cloud cover also requires amendment to the image selection decisions in Fig. 1, such that image cloud cover and look angle are considered less important. However, the availability of imagery in Google Earth remains critical, and the order of importance of the spectral resolution, spatial resolution, and swath widths remain unchanged. In arid environments, the occurrence of landslides may be less detectable by spectral changes to the land surface than by morphological changes. A judgement may therefore be required as to the relative importance of image spectral and spatial resolution.

## 4.2 Can manual landslide mapping provide useful information quickly enough to inform humanitarian response efforts?

Generating a useful assessment of landsliding immediately after an earthquake remains challenging due to a lack of clarity around what information is possible to acquire under severe time constraints, and what information is actually useful (Robinson et al., 2017). Our mapping effort showed that delays in information production can occur due to: image availability, image quality, cloud cover, and the time taken to handle and map from imagery once it became available. While some clarity on increasing the speed of these processes can be provided via reflections such as this, pertinent information is inevitably unique to each earthquake and its socio-political context. At the highest level, information on landsliding within the first 72 hours can help to define the scale, extent, and distribution of landslide impacts across the entire affected area, particularly if this area is otherwise inaccessible. Given the delays in image capture and mapping, full landslide mapping for an event on the scale of the Gorkha earthquake or larger is impossible to achieve within this 72-hour timeframe. However, as the number and exact location of all landslides is not important to disaster managers at this stage of a response (OCHA, 2013; IASC, 2015), a faster approach is preferable.

Robinson et al. (2017) explored the merits of seeding an empirical landslide model with the initial outputs from rapid post-earthquake mapping efforts, such as our initial attempts (Fig. 3). They found that small numbers ($\sim10^2$) of mapped landslides were sufficient to accurately predict the spatial hazard posed by $\sim10^4$ landslides as long as their distribution covered a large portion of the affected area. Here we have shown that such small numbers of landslides can be mapped within the 72-hour timeframe. Importantly, however, when models and empirical data are presented together, their relative merits and drawbacks need to be clearly articulated. For example, while models can suggest where landsliding is more or less likely to have occurred with varying degrees of certainty, direct observations provide absolute certainty at some locations, but remain inherently uncertain where the ground has not been observed. Conversely, combining models and observations to draw conclusions about the likely presence of landslides where the ground has yet to be observed may enable faster dissemination of information to end-users where full mapping is not practicable. Using gaps in the initial cloud cover, our identification of valleys of severe landsliding and prediction of the southernmost extent of landsliding was achieved within two days of images becoming available. This highlights the importance of nested monitoring within SEM (Voigt et al., 2016) whereby coarser imagery with large footprints can be used to identify areas of concern, which can be subsequently monitored using higher resolution approaches.

A clear exception to this finding is in assessing the imminent potential for secondary hazards posed by landslide dams (e.g. Cui et al., 2009; Kargel et al., 2016). It is widely recognized that landslide dams typically fail soon after formation, with 41% failing within one week (Costa and Schuster, 1987). Rapid assessment to inform the management of this risk is therefore vital. However, features indicative of progressive failure, such as widening tension cracks, are too small to be visible in even the highest resolution satellite imagery, and so SEM is mostly valuable for locating and low-resolution monitoring of landslide dams. An appraisal of the risk that they pose is best undertaken on the ground.

Our findings suggest that there is potential additional value in informing post-earthquake landslide mapping efforts to target medium to longer-term information needs, as well as the immediate response. The transition from disaster response to recovery can occur over a matter of days, and while some information gathered in the immediate earthquake aftermath may not be instantly useful, it may become valuable for later decision. For example, given that earthquakes elevate landslide hazard for sustained periods of time (e.g. Marc et al., 2015), continually updating coseismic landslide maps to assess how the hazard evolves is potentially of great value, yet is rarely undertaken. In the aftermath of the Nepal earthquake, there were 46 days between the mainshock and the first rainfall-induced fatal landslide of the monsoon. Detailed mapping that describes individual landslides and the potential for remobilization is invaluable in assessing risks during future monsoons. However, as such uses require a high level of local detail and precision, mapping must be accurate, which can be difficult to achieve within limited timeframes. Defining the aim and output of responsive mapping is therefore vital to establish the data that must be collected. It is equally clear that there is no requirement to wait until an earthquake occurs to start defining what information could be useful with those charged with managing the response. Scenarios or planning exercises are widely used to prepare those involved in disaster response (Davies et al., 2015), and could be extended to consider coseismic landslide hazard assessments, to define what information can be provided and when. This process would be of value to end-users, but also to those producing landslide assessments to ensure that aims are realistic and defined by needs. Similar discussions for other forms of geohazard have benefitted from protocols and guidelines that aim to standardize approaches, outputs and procedures (UN-SPIDER, 2015). Groups such as the CEOS Working Group on Disasters, and the UN-SPIDER IWG-SME, are vital frameworks for establishing these technical, practical and ethical guidelines on SEM for coseismic landslide assessment.

### 4.3 The best way to map coseismic landslides?

In circumstances where mapping individual landslides is of value, the choice of whether to digitize points, polylines or polygons is an important consideration. The choice must be based on the extent of the mapping area, the time available for mapping, and the number of landslides to map. However, estimating the number and extent of landslides in the immediate aftermath of a disaster is complex, and the choice of digitisation technique must be open to change in response to reasonable assumptions about the nature of the event. This decision is also based on the desired outputs and the scale at which they will be used. The reliability of the geometrical data provided by polygons, while beneficial, is highly sensitive to the accuracy and consistency of image orthorectification, which are challenging in steep terrain. We observed that, where a landslide spanned an altitudinal range of more than several hundred metres, the accuracy of results generated strongly depended upon the spatial resolution of the imagery and the sensor incidence angle. As a result, where multiple data sources are used and image resolution varies across the affected area, the number and size distributions of polygons also vary, leading to systematic inconsistencies in mapping. Coarser, and hence more rapid, methods of mapping are valuable for a rapid assessment of landslide impact across the whole earthquake affected area, but are less useful for understanding individual landslides. We found that polylines offered a compromise that retains some of the speed of mapping points, but also enables an assessment of landslide size and intersection with features of interest, such as roads, buildings, or rivers.

Semi-automated and automated approaches to image segmentation hold potential for more time efficient landslide mapping, with considerable success reported outside immediate post-disaster contexts (e.g. Tsai et al., 2010). However, discernible spectral changes across a landscape, upon which pixel-based segmentation depends, may only occur for failures within densely vegetated areas that have the potential to revegetate over short periods. A reliance upon spectral responses can also result in the misclassification of channel bank erosion and fluvial sedimentation, the misidentification of reactivations, and the division of large landslides into multiple fractions. While the increasing availability of VHR imagery directly enhances the accuracy of manual landslide mapping, the results of automated and semi-automated pixel-based methods that have used VHR imagery are susceptible to large spectral variance between pixels, creating intra-class variability, and are more sensitive to coregistration errors (Moine et al., 2009; Martha et al., 2010; Mondini et al., 2011). Object-based image analysis overcomes many of these issues by accounting for additional metrics such as color, texture, shape and topography (Stumpf and Kerle, 2011), though the selection of useful object metrics is time intensive and varies from case to case. Both approaches are likely to benefit from the rich spectral information gathered by medium resolution sensors, such as Sentinel-2, and short revisit periods that enable access to pre-event datasets. However, while the speed gain of (semi-)automated methods over manual methods increases with the area to be mapped, larger areas also increase the reliance upon imagery from a variety of sensors. The application of semi-automated and automated mapping with variable image characteristics and quality is yet to be reported. Future research into the use of Sentinel-2 imagery is therefore required (Voigt et al., 2016), and these approaches may yield an important assessment that sits between landslide probability models and manual landslide mapping from optical imagery in the aftermath of a trigger event (e.g. Stumpf et al., 2017).

In instances where cloud cover is prominent, the use of satellite-borne radar also has the potential to provide an assessment of large landslides prior to mapping from optical imagery. Large failures may be rapidly identified by significant morphological changes, such as shifts in the channel network. Alternatively, a large-scale shift in the dielectric constant of the slope, as vegetation is removed, may be detected by changes to the amplitude of the backscattered waves (Jin et al., 2009; Mondini et al., 2017). In this manner, SAR amplitude/intensity images have been used to map single landslides at the slope scale (Raspini et al., 2015; Plank et al., 2016) and, more recently, at the catchment scale following triggering events (Casagli et al., 2016; Mondini et al., 2017). However, SAR imagery requires a considerable amount of complex pre-processing and the accuracy of change is highly sensitive to the image acquisition geometry, which can be sub-optimal in mountainous regions.

## 4.4 What limits the time needed to produce a useful landslide assessment?

The time taken to produce outputs from our mapping campaign was most influenced by image availability, specifically that which was cloud free over the area of interest. For this earthquake, the workload of five analysts appeared to yield a suitable balance between capacity, shared learning, and consistency, given the timeframes to produce outputs. It was beneficial for all mappers to be in one laboratory, enabling easy coordination and communication to ensure coverage and consistency and to avoid replication. We were able to partition the earthquake-affected area into regions of interest for each mapper, and these regions were dynamically updated in response to the availability of high(er) quality imagery. Given the increased capacity of

the SEM community to develop map products in recent years, this partitioning represents an important phase in the coordination of multiple groups, thereby avoiding repetition and increasing the consistency of outputs (Voigt et al., 2016).

The introduction of larger satellite constellations with more advanced sensors also expedites the availability of imagery for future mapping campaigns, increasing the efficiency of post-disaster mapping (Voigt et al., 2016). For example, Sentinel-2
combines a large swath-width (290 km) with a moderately high spatial resolution (10 m visible and near-infrared), which will reduce the number of images, and thus processing time, required to cover large areas. In addition, the shorter return period (five days for Sentinel-2a and -2b, compared to 16 days for Landsat 8) will increase the probability of observing the ground through gaps in any cloud cover, reducing the time needed to produce outputs. Our effort demonstrated that once imagery is available, mapping can be rapid (two to three days), given suitable capacity. However, we have also found that it cannot be
assumed that a landslide inventory or assessment will be possible to generate immediately, once an image is captured. This is a problematic assumption that raises expectations of both those producing landslide assessments, but also those who could use them.

The timeliness of an SEM landslide assessment must be considered relative to alternative sources of information. While each earthquake is different, multiple sources of information will become available to decision makers, primarily based upon
networks collating human intelligence from those on the ground. In Nepal, nationwide systems capable of rapidly assessing the earthquake impacts included the networks of the military, Red Cross, and local government. Such approaches can, however, be subjective, incomplete and inconsistent in coverage, and cumbersome to administer (OCHA, 2013; Datta et al., forthcoming). Inevitably, such assessments are also restricted to areas with functioning communications or to accessible parts of the road network, at least until systematic reconnaissance can be undertaken. Such systematic reconnaissance is also highly
contingent upon favourable weather and available resources. Consequently, some areas can remain isolated for days or weeks. For example, the Jhelum Valley in Pakistan after the 2005 Kashmir earthquake (Petley et al., 2006; Owen et al., 2008; Mahmood et al., 2015) and the Rasuwa and Upper Bhote Kosi valleys after the 2015 Nepal earthquakes were left isolated by landsliding, leaving the status of thousands of households largely unknown as the wider response effort gained pace.

## 4.5 Science, citizen science, and coseismic landslide assessment

Through the proliferation of mobile technologies, open-source mapping, and online GIS, an increasingly important role for social media and crowd-sourced data in disaster response is emerging (e.g. Zook et al., 2010; Fleischhauer et al., 2017). Following the Gorkha earthquake, crowd-sourced mapping campaigns initiated by Tomnod (with imagery from DigitalGlobe™) and OpenStreetMap (with imagery from Airbus) provided users with access to image tiles and the ability to create and edit vectorized shapes. These sites produced damage maps that were used extensively by the Nepali military, both
for logistics planning and for identifying communities in need of assistance (The Nepalese Army, 2015). The value of such crowd-sourced information has also been recognized by the scientific community in response to several recent natural disasters (e.g. Goodchild and Glennon, 2010; Barrington et al., 2012; Roche et al., 2013; Poiani et al., 2016).

To date crowd sourcing has not, however, been employed to map coseismic landslides in a manner that is reliable. Landslide mapping requires pre- and post-earthquake datasets, knowledge of failure processes and mechanics, and an understanding of what is possible to observe based on the spectral characteristics of the imagery. Research is needed into how best to support crowd-sourced mapping to generate reliable landslide mapping and inventories, and to feed learning from compiling science-

focussed landslide inventories into this process. In our campaign, we also benefited from insights from social media to identify and locate landslides in areas with persistent cloud cover. A combination of archived pre-earthquake imagery and reported locations allowed us to locate the exact hillslope that had failed in 20 locations, the positions of which were later verified by our formal mapping. A platform that permits this combination of data with more conventional mapping therefore offers an attractive means of collating and verifying landslide data.

Advances in collating landslide inventories, including crowd sourcing, and the key messages that can be distilled from their analysis, are valuable for disaster response. However, key messages need to be articulated quickly and clearly along with any associated limitations or uncertainties. The various means of landslide assessment that have been discussed above are summarized in Table 2. This provides a chronology of outputs that clarifies what we have found possible to achieve within the timeframes of the UN Situation Analysis and MIRA report. The various means of landslide assessment that have been

discussed are summarised in Table 2. This provides a chronology of outputs that clarifies what we have found possible to achieve within the timeframes of the UN Situation Analysis and MIRA report. The timescales of what is possible will vary between events, predominantly as a function of cloud cover for landslide mapping, but the suggested timescales in Table 2 are broadly independent of this. For example, following the first cloud-free imagery after the Gorkha earthquake, the production of an initial landslide assessment and inventory was available within approximately five days, as reflected in the description

of a full point inventory. The benefits and limitations of each are included to provide detail on what is and is not possible to conclude. Importantly, once a dataset is made available online, it is publicly available for the foreseeable future. While this provides a good base for others to work from, care is needed in how and where data are shared and how caveats and uncertainties are communicated, in particular the method used to generate the dataset. Based on our experience of communicating landslide assessments, each published output requires the following accompanying information: (1) a

supporting narrative that describes the aims, assumptions, methods, and limitations of the data; (2) a high-level analysis of the key messages or conclusions that can and cannot be reached on the basis of the mapping; (3) a statement of intent for further work, so that end-users can see how the work will evolve; and (4) a mechanism for feedback or exchange between mappers and end-users. Unless these elements are made available, the output is likely to be either overlooked, or it may be used in ways which were not intended.

**4.6 Recommended approach to manual mapping using optical imagery**

Based on our experiences of the 2015 Nepal earthquakes, we provide the following recommended approach to manual mapping of large numbers ($> 10^2$) of landslides in the aftermath of a trigger event. As discussed in Section 4.1, this approach will vary

based on the density of landsliding, the area to be mapped, the number of mappers available, image acquisition timing, and cloud cover.

- Choosing the best imagery, which has sufficient spectral and spatial resolution, minimal topographic distortion and continuous spatial coverage, is a key primary consideration prior to mapping. The area that has suffered shaking sufficient to trigger landsliding (> ~$M_W$ 4.0; Keefer, 1984) should first be identified using initial outputs from USGS ShakeMap. Likely data availability arising from future satellite overpasses of this area should be assessed along with weather conditions to determine the extent of cloud cover. This catalogue is essential to plan the likely timescales involved in completing various stages of mapping, and should be attained within several hours of the event.

- There are significant gains to be made by combining manual mapping and empirical modelling of coseismic landsliding. Within the first 24 h, the outputs from empirical models are likely to provide a useful indication of the area impacted by landsliding, which can be used to guide subsequent mapping efforts. Such models can be verified relatively quickly by manually delimiting the area impacted by landsliding, without a need to map each individual failure. These should be examined alongside Copernicus Emergency Management Service reference maps. An online search for documented landsliding on the ground also provides useful information for targeting individual slopes. These information sources are particularly useful in locations where the mappers have no background knowledge of landsliding or baseline datasets, and should be examined within 48 hours of the earthquake.

- Pre-event imagery must be sought to ensure that only landslides triggered by the event, or those remobilised, are mapped. Medium resolution (Sentinel-2 or Landsat 8) imagery is sufficient as a baseline dataset. High resolution imagery made available in Google Earth may also prove useful, as long as the most recent image acquisition occurred after previous regional meteorological events, such as the South Asian monsoon.

- Preliminary outputs, which precede a full inventory and can be produced much more quickly, can be of value to disaster managers on the ground. This includes the locations of valley blocking events, areas of severe landsliding, and other general observations. Where available, high resolution imagery from tasked sensors should be used in the first instance in order to identify valley blocking events as each image tile is made available. However, given that the initial focus of such imagery is likely to be over urban centers and the epicentre, it cannot be assumed that these first datasets will cover the total area affected by landsliding. Once medium resolution imagery covering a larger area is made available, this can be used to manually identify valley blocking events over the entire area within hours. Only once a valley blocking event has been breached should its monitoring be discontinued.

- Areas of severe landsliding should be noted during searches for valley blocking events. Details of the most severely affected valleys and the approximate region affected by landsliding should be quickly disseminated. This need not necessarily constitute a formal map product.

- Selecting the most suitable mapping platforms needs to weigh the speed of access to data against the ease with which mapping can be undertaken. Once the above stages are complete, formal individual landslide mapping can begin. The mapping platforms available should be assessed and a consistent protocol established amongst those involved. If

imagery is available through platforms such as Google Crisis, these have the advantage of removing the need for imagery download and processing, but can mean delays in obtaining access to the latest imagery. Such platforms also allow pre- and post-event imagery to be compared, and overlaid with a terrain model.

- The chosen mapping method has a significant impact on the time needed to map large numbers of landslides. If time is limited, mapping landslides as points is advantageous. A map of landslide points, significantly affected valleys, and the area within which points are found should be possible within one to three days of the first medium resolution imagery. This is equivalent to the creation of Copernicus Emergency Management Service delineation maps, which provide an assessment of the extent of the event.

- The highest resolution data may not always be the most appropriate for wide-area mapping of landslides. From our experience, medium resolution imagery such as Sentinel-2 and Landsat 8 currently provides a good balance between image footprint size and coverage, and spatial and spectral resolution. Senitnel-2 imagery has a frequent return interval (five days), increasing the probability of image availability in the days after an event and providing recent pre-event imagery. High-resolution imagery, which tends to have a smaller footprint, is best if incidence angles are close to nadir, such as < 20°, to avoid time-consuming georeferencing. An exception to this applies for monitoring of identified valley-blocking landslides.

- Outputs should be open access and clearly explained. Maps should be made available in open formats, alongside a description of the methods, limitations, and key messages. Accompanying vector data should also be provided given that the value of much of this data is when it can be overlaid with other data, such as assets or infrastructure. If possible, feedback on the data being produced from those using it on the ground is valuable.

- If there is a continued need to generate more granular detail, landslides should be individually delineated using polylines, as a compromise between speed and detail as compared to points and polygons. Polylines enable the magnitude of events to be approximated and can be used in combination with infrastructural data in order to identify events that may have caused highway blockage or damage. In some developing regions, vector data is likely to improve with time following the event due to crowd-sourced mapping initiatives. Polyline mapping of the area is potentially possible to complete within approximately one week and a map product provided. Maps of the number of landslides per unit area (density of landsliding) are useful indictors of the extent and spatial distribution of relative landslide intensities, and any accompanying landslide vector data should be made available.

- Polygons are only recommended for mapping landslides if capacity permits and where imagery is suitable. Where imagery is subject to high levels of topographic distortion and therefore poor registration, there is little gain in meticulously mapping landslide extents with polygons, both from a scientific and from a risk reduction perspective. The time required to produce this data is also highly likely to exceed the timeframe within which it is needed to inform the initial disaster response. Small numbers of landslides mapped with polygons distributed across the area delineated in the initial point-based mapping could become useful as training datasets for landslide probability models and

automated mapping (e.g. Stumpf et al., 2017). In such instances, this mapping should occur in parallel to all other mapping.

## 5 Conclusions

In this paper, we have reflected on our experience of creating an inventory of coseismic landslides rapidly after the 2015 Nepal earthquakes. While scientific efforts to map coseismic landslides may aim to assess the hazard in an urgent manner to inform the humanitarian response, they are rarely completed rapidly enough to do so. As such, scientific efforts to generate useful information require recognition of what is both useful and practicable within the available timeframe. We have demonstrated what can realistically be achieved, including the time critical decisions that need to be taken to expedite the mapping process. While any increase in the rate of image availability increases the likelihood of producing useful landslide assessments, the consideration of what is possible (given handling and processing constraints on mapping) and what is useful (given the priorities of end-users responding to humanitarian crises) remains pertinent for other future events.

Our lessons can and should inform the approach and expectations of those who seek to produce rapid (days to months) coseismic landslide assessments, and those who would benefit from using this information. There is clearly no requirement to wait until an earthquake occurs to begin conversations around what is or could be useful, and these conversations should involve scientists, government representatives, and humanitarian response teams. The efforts of UN-SPIDER and the CEOS Disaster Working Group are vital for ensuring coherence in the response to future earthquakes. With rapid advances in social media and accessible geospatial data, it is likely that future post-earthquake assessment will benefit from more systematic crowd-sourced data collection and integration.

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

**Tables**

---

**Table 1: Example of notes that accompanied the map released on 18th June, an extract from which is presented in Fig. 6.**

---

The Dolakha aftershock on 12 May (15 days after the 25 April mainshock) prompted a second campaign of mapping in response to reports of further landslides close to its epicenter. In the week following 12 May, the majority of new optical imagery was acquired with incidence angles of 25-45°. Given the extreme relief in the epicentral region, it was decided to delay mapping until imagery that was more suitable became available. A landslide map derived from imagery collected after both earthquakes was therefore not released until 18 June (54 days after the mainshock), with landslides categorised as follows:

(1)    Failures positively identified as occurring as a result of the 25 April Gorkha earthquake

(2)    Failures positively identified as occurring as a result of the 12 May Dolakha earthquake

(3)    Failures that occurred *either* as a result of the 25 April Gorkha earthquake *or* 12 May Dolakha earthquake, having occurred in areas where cloud-free imagery was only available after 12 May.

(4)    Failures considered likely to have been caused by either the Gorkha *or* Dolakha earthquake, but where pre-earthquake imagery was only available prior to the 2014 monsoon season.

(5)    Landslides that had been observed after the 25 April Gorkha earthquake but which had not changed after the 12 May Dolakha earthquake

---

**Table 2: Timescales, benefits, and limitations of landslide-related outputs, based on response to a large continental earthquake in a mountainous region. Approximate timings described are based on experience of undertaking landslide assessment after the 2015 Gorkha earthquakes, and related studies, but will inevitably vary between events.**

| Output | Timescale | Benefits for landslide assessment | Limitations for landslide assessment |
|---|---|---|---|
| Epicenter location, depth, and local magnitude | Seconds - minutes | Rapid event location and scale (magnitude). Earthquake magnitude and depth broadly relates to the scale of landslide impacts, based on e.g. Keefer (1984). | Single point location, rather than impact footprint. Empirical links between earthquake magnitude and landslide impacts have ~2 to 3 orders of magnitude of uncertainty, and so preliminary assessments are reliant upon expert judgement. Earthquakes rarely have local, directly comparable precedents, and the spatial distribution of landslides that they trigger is based upon multiple characteristics of the rupture (e.g. area and depth) as well as the overlying topography. Typically focusses response attention to the epicenter, which may not be the most in need. |
| Modeled shaking intensity, e.g. USGS Shakemap | < 1 h onwards | Identification of area affected by shaking. Can steer relief focus to wider impacted area rather than just epicenter. | Model does not directly predict landsliding, but assumes some correlation between shaking intensity and landslide occurrence. Model is reliant on availability of instrumental records, and is continually updated and refined as new data becomes available. Final version may not be available until weeks-months after the earthquake. |
| Aerial reconnaissance (e.g. military, expert) | < 1 h onwards | Initial flights, commonly by military, over the affected area can provide a 'first look' assessment of the nature and scale of landslide impacts. Can put limits on the landsliding extent and intensity along the flight track. Systematic flights may follow, enabling more targeted and extensive coverage (e.g. USGS / GEER response described in Collins and Jibson, 2015), as well as analysis of failure evolution / reactivation if an area is revisited. | The route for flights is weather and resource dependent, and may be directed by only limited data, such as the epicenter location. For large earthquakes, complete systematic reconnaissance of the affected area is challenging, and it is unlikely that protocols for mapping impacts are in place at this time. Landslide assessment is unlikely to be the sole purpose of such initial flights, and so systematic data collection is unlikely. Accuracy in locating impacts not directly beneath the flight path can be limited. |

| | | | |
|---|---|---|---|
| Empirically modeled earthquake-triggered landslide maps | < 24 h onwards | Models capable of predicting spatial probability of landslides, footprint and relative intensity of impacts, size distribution, runout, and impact on buildings and infrastructure. Can feed into 72 h Situation Analysis timeframe, and can direct efforts for more detailed assessment. Modelling is independent of weather that might otherwise restrict aerial reconnaissance. Potential to run models in near real-time with ShakeMap. | Heavily reliant on: (1) quality of modeled shaking intensities and availability of instrumental records; (2) availability of input data (e.g. topography, assets); and (3) assumes model training data is sufficient to predict event specific characteristics in hand. Models do not predict individual landslide locations and only provide relative impacts or probabilities, which can be difficult to communicate or interpret. |
| Social media and crowd-sourced information (e.g. Goodchild and Glennon 2010; Barrington et al. 2011; Roche et al. 2013; Poiani et al. 2016). | 1 h onwards | Inevitable focus on immediate impacts on population and infrastructure, which is largely unaffected by weather. Can be very agile, with increasing coverage even in remote areas. | Quality control is challenging to enforce as reports are subjective, and locations can be difficult to ascertain. Reliant on functioning communications. Potential bias towards populations / infrastructure restricts ascertainment of total spatial extent and relative intensity of damage, and does not consider more remote latent hazards such as landslide dams. Qualitative local assessments are difficult to extrapolate to relative measures of impact. Critically, no report does not mean no impact. |
| Polygon of landslide impacts from satellite imagery and estimates of relative intensity of landslide affected areas | First available imagery* + 6 h | Direct positive identification of spatial extent of landslide impacts from optical satellite data captured after the earthquake, where mapping individual landslides is not required to delimit the extent of impacts. Valuable for informing response logistics, and imagery itself provides understandable map of impacts. Can be achieved with medium-resolution imagery (e.g. Landsat). Identification of intense impacts informs location of initial relief delivery and airborne assessments. Can be assessed qualitatively without the need for full coverage with each individual landslide identified. | Reliant on cloud-free imagery. Lighting, vegetation cover, and steep topography may make interpretation challenging. Landsliding may have poor radiometric contrast with unbroken ground, making landsliding difficult to identify. Subjective definition of severity remains. |
| **Landslide mapping**<br><br>Automatic landslide mapping (e.g.; Martha et al., 2010; Mondini et al., 2011; Lu et al., 2011; Đurić et al., 2017; Hölbling et al., 2017) | First available imagery* + 1 h** | Potentially rapid generation of a polygon-based landslide inventory across the entire affected area. Time to complete is expected to reduce as technology improves and more experience is gained (Voigt et al., 2016). | Technique still in infancy, restricted to cloud free optical imagery, and reliant upon a style of landsliding that is readily visible in post-earthquake imagery. Use of SAR for rapid inventory generation, in particular through cloud cover, is still in its infancy (e.g. Casagli et al., 2016). |

| ↑ | ↑ | ↑ | **Products can inform the 72 h Situation Analysis** | ↑ | ↑ | ↑ |
|---|---|---|---|---|---|---|

| | | | | | |
|---|---|---|---|---|---|
| Site-specific landslide dam assessment (e.g. Kargel et al., 2016) | First available imagery* + ~3 days onwards | Requires assessment of possible dam locations in the fluvial network across the impacted area and inspection in imagery of known newly formed dams, but critical to complete as quickly as possible after earthquake to mitigate downstream secondary impacts. Not reliant on having a landslide inventory of full coverage. | Dependent on high-resolution cloud free imagery. Ideally benefits from time-series imagery to assess dam evolution and stability, without combining upstream and downstream flow monitoring. Future landslide dam stability remains difficult to forecast. | | |
| Landslide mapping: Full coverage (points) (e.g. Kargel et al., 2016) | First available imagery* + ~5 days | Relatively quick to create an inventory of full coverage. Enables locations with intense damage to be identified (and hence e.g. roads liable to blockage), and relative intensity of impacts across the affected area to be quantified to guide response resources, and area of landslide impacts to be positively identified. Minimal sensitivity to orthorectification errors in steep topography. | Cloud cover and image acquisition dependent. Limited ability to appraise landslide size, mechanism and risk. Proximity to and impact of landslide on infrastructure difficult to appraise. Landslide numbers alone of questionable value to responders. Hard to manage consistency, and commonly has to be mapped from multiple images from multiple sources, making mapping potentially inconsistent. | | |
| Landslide mapping: Full coverage (polylines) (e.g. This study) | First full available imagery* + ~7 days | Relatively quick to create compared to polygons, and requires only slightly more effort relative to points. Polyline length can act as proxy for landslide size, and enables appraisal of landslide proximity to population and infrastructure, providing a binary assessment of location of impacts. Mechanism can be inferred from polyline length (e.g. debris flow v slump). | As above, but estimates of landslide size are potentially susceptible to image distortion particularly in steep topography and off-nadir satellite view directions. Sensitive to instances in which cloud cover partially obscured the landslide track. | | |
| Landslide mapping: Full coverage (polygons) (e.g. Martha et al., 2016; Roback et al., 2017) | First full available imagery* + 2 weeks minimum | Area and volume estimation for scientific use. Allow analysis of future change to landslides (e.g. post-monsoon). Enables full appraisal of landslide proximity to population and infrastructure, and assessment of potential magnitude of impacts. | As above. Arguably limited added benefit for relief effort as compared to polyline mapping. Relatively slow to generate. Highly sensitive to orthorectification and georeferencing errors. | | |
| ↑ | ↑ | ↑ | **Products can inform the two-week MIRA report** | ↑ | ↑ | ↑ |

*Refers to the latency of cloud-free image acquisition (typically ~24 – 72 h), the duration of which is likely to vary considerably in mountainous regions.

** Estimated duration of automated landslide mapping currently unknown

**Figures**

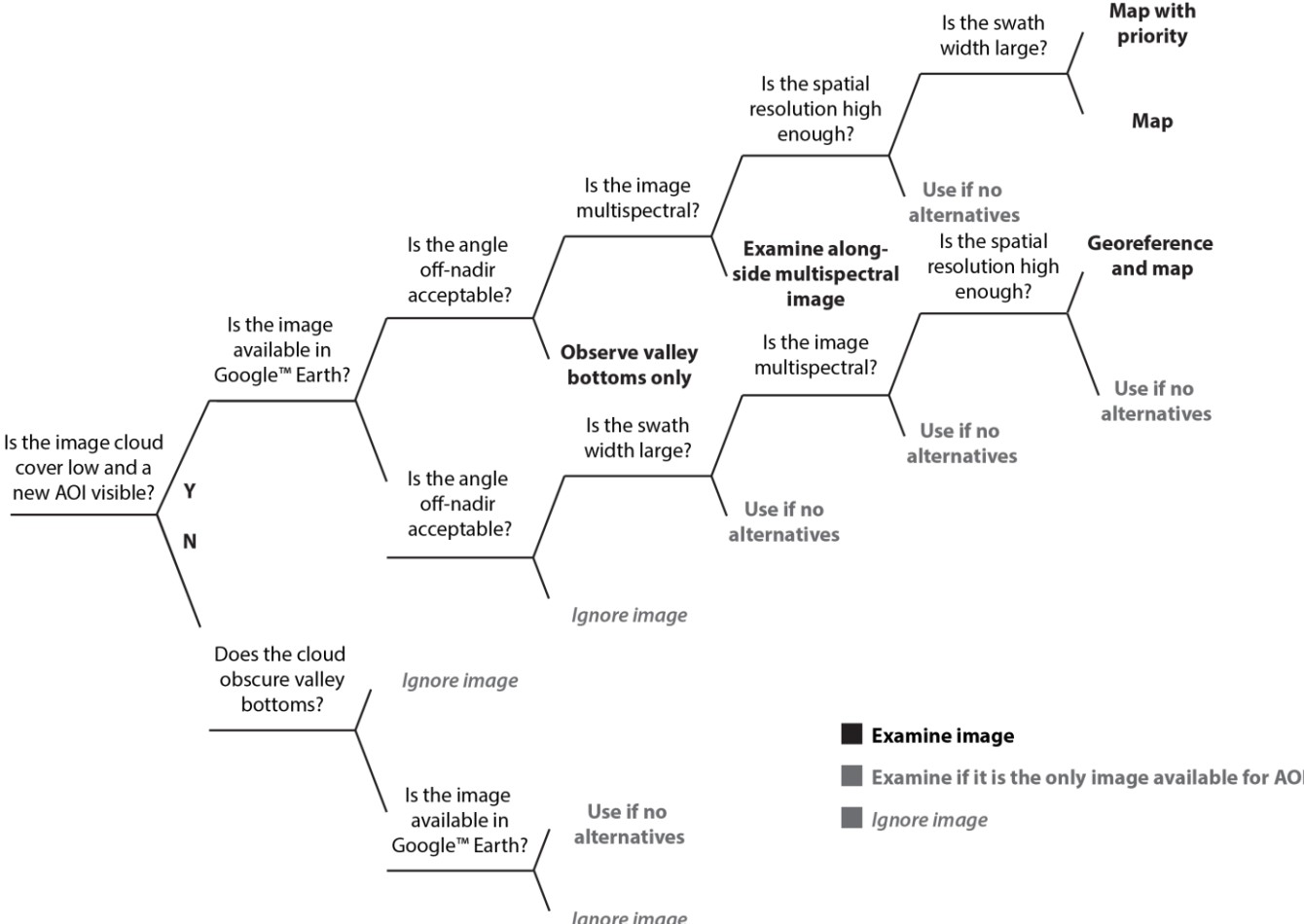

Figure 1. Decision tree for prioritising imagery used by Durham University for landslide mapping after the 2015 Gorkha earthquake. The relative importance of criteria decreases from left to right. Datasets were prioritized if they were efficient to pre-process and provided high-resolution data are optimal for mapping. Imagery with large swath widths and acceptable off-nadir angles may be difficult to acquire in mountainous terrain. These criteria were therefore prioritized to reduce the time spent georeferencing and the number of images required. Given the sub-metre resolution of VHR imagery and the ability to pan-sharpen multispectral imagery, most image resolutions are now sufficient to map landslides with the potential to cause significant damage. Spectral resolution was therefore considered as a more useful criterion for distinguishing landslides of this type than spatial resolution. This decision tree may also be applied to image selection for automated landslide mapping.

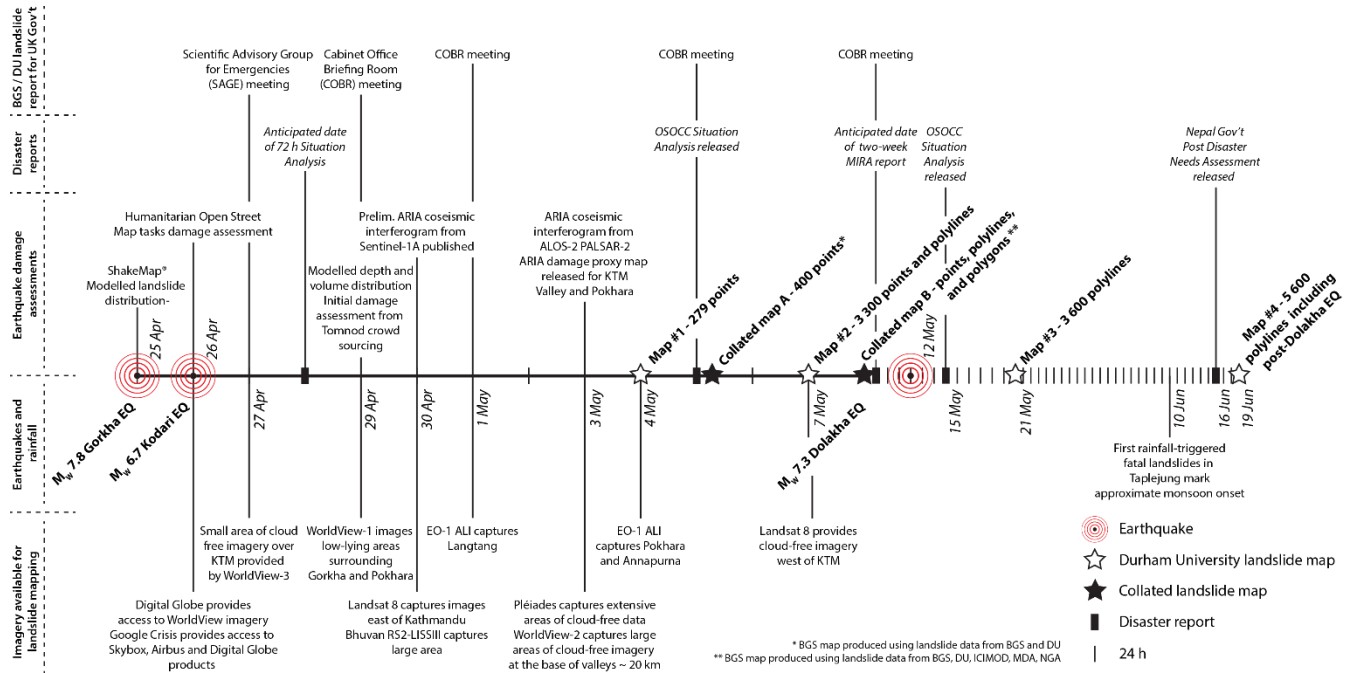

**Figure 2: Timeline of image acquisition, mapping, disaster reports, and other earthquake damage assessments from 25 April 2015. Earthquake timing is also added alongside the approximate onset of the monsoon on 10 June (46 days after the Gorkha earthquake). The timing of OCHA On-site Operations Coordination Centre (OSOCC) Situation Analysis reports and the Nepal Government's Post Disaster Needs Assessment (PDNA) are added alongside the proposed timings of the Situation Analysis and MIRA report as defined by IASC (2015). No MIRA report was created following the Nepal earthquakes due to logistical difficulties in organising its creation and physical access constraints (ECHO, 2015). The timeline is non-linear, with each vertical line representing one day.**

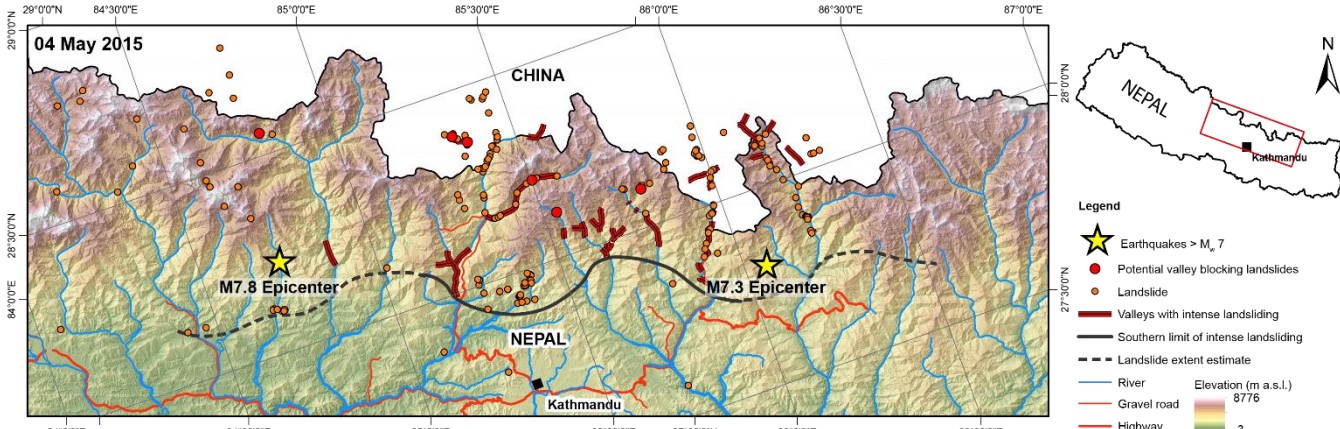

**Figure 3: Extract from landslide impacts map released on 4 May 2015, nine days after the Gorkha earthquake and two days after cloud cover recession. Orange dots represent the location of observed individual landslides, at the point at which they reached the valley base. Red dots represent potential valley blocking landslides that had the potential to inhibit river flow, posing a future breach risk downstream. Red lines represent valleys identified as having experienced very intense landsliding, predominantly rockfall and dry debris flows. The black line delimited the southern limit of the area of intense landsliding. This limit was observed where solid and was anticipated where dashed, given that it was not visible in imagery. Both the 25 April (Gorkha) and 12 May (Dolakha) epicenters are added to this map for reference, despite its release prior to the Dolakha earthquake.**

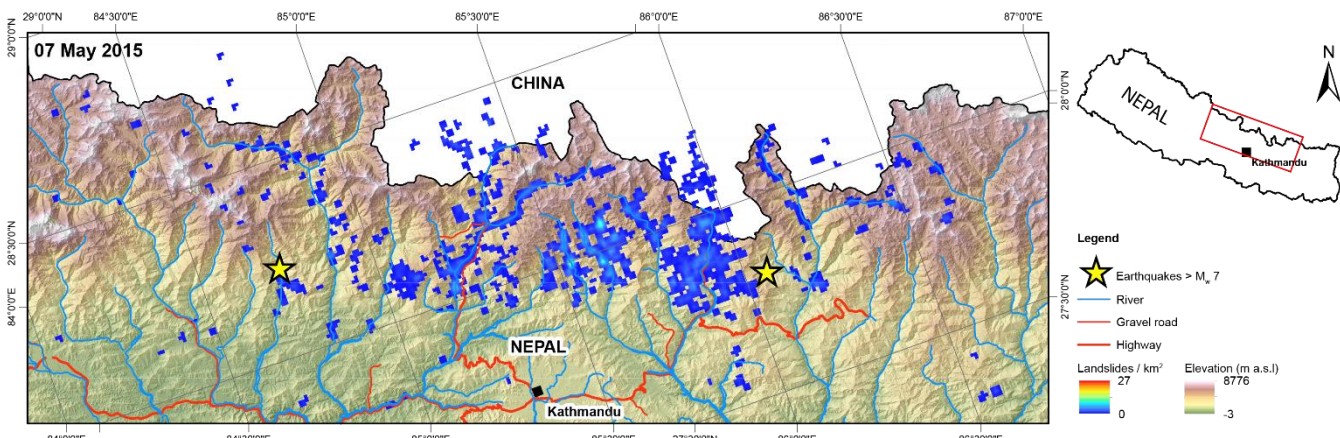

**Figure 4: Extract from map released on 7 May 2015, 12 days after the Gorkha earthquake. Colored zone shows landslide distribution and relative intensity (number of landslides / km²). The colour map has been adjusted to a range of 0-27 landslides / km² for comparison between Fig. 4-5. At this point, all areas in the map extent had been assessed using at least pan-sharpened Landsat 8 imagery (15 m). VHR (< 3 m) optical imagery had been used where available.**

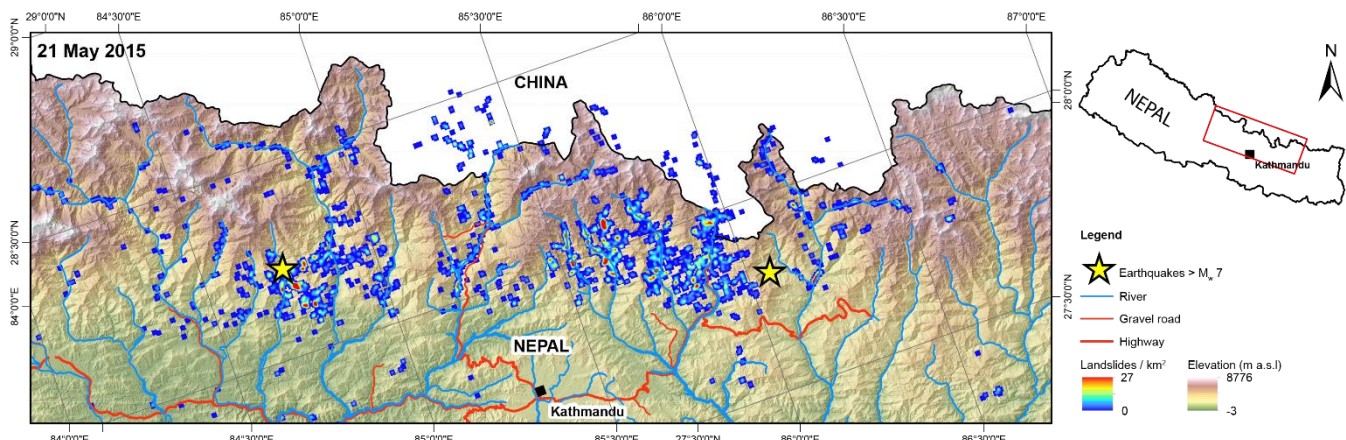

**Figure 5: Extract from map released on 21 May, nine days after the Dolakha earthquake. Due to cloud cover and image acquisition, this map did not include landslides that occurred following the Dolakha earthquake.**

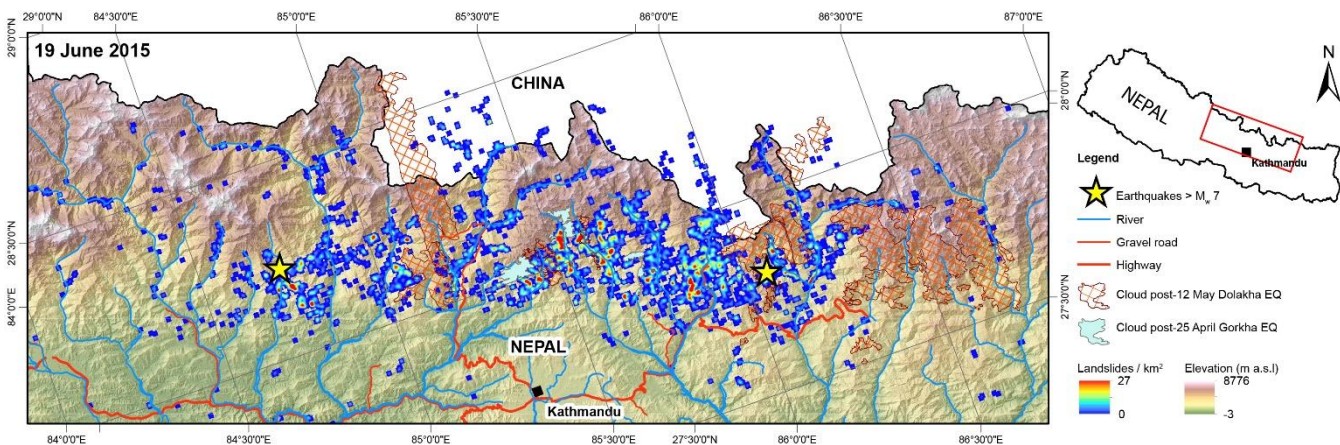

5 **Figure 6: Extract from map released on 19 June 2015 containing landslide data from both earthquakes, comprising ~4 500 triggered by the Gorkha event, ~300 by the Dolakha event, and ~800 that could be attributed to either. Orange hatched pattern highlights areas that could not be mapped following the Dolakha earthquake event. Turquoise pattern (direct north of Kathmandu) highlight areas that remained unmapped following both earthquakes.**

**Acknowledgements**

The research was funded by NERC Urgency Grant NE/N007689/1, the NERC-ESRC 'Earthquakes without Frontiers project (NE/J01995X/1), GCRF Grant NE/P016014/1, and the UK Department for International Development (DFID) as part of the Science for Humanitarian Emergencies and Resilience (SHEAR) program. This study has also been in part supported by the
5    DIFeREns2 (2014-2019) COFUND scheme supported by the European Union's Seventh Framework Programme (#609412). We are grateful to H. Bell and S. Whadcoat from Durham University who provided assistance during the mapping. We thank the various agencies that made satellite imagery freely available through the International Charter on Space and Major Disasters. We also thank the United Nations Office for Coordination of Humanitarian Affairs (UN-OCHA) and the UN Resident Coordinator's Office in Kathmandu, and DFID offices in London and Nepal, whose input in helping to define the
10   timelines of useful information for emergency response greatly informed this study. C. Jordan and T. Dijkstra publish with the permission of the Executive Director of the British Geological Survey.