# Peer review of "Satellite-based emergency mapping: Landslides triggered by the 2015 Nepal earthquake"

_Natural Hazards and Earth System Sciences, 2017_

## Referee Comment (RC1) · O. Marc (Referee) · 25 Aug 2017

Review of "Satellite-based emergency mapping: Landslides triggered by the 2015 Nepal earthquake" by William et al.

**General summary:**

The paper by William, et al., reports on the experience of the authors on mapping and publishing emergency maps odof earthquake-induced landslide after the Gorkha earthquake in Nepal 2015. They summarize the challenges associated with the effort of delivering products useful to humanitarian practitionner and propose and discuss approaches to tackle them.

Although limited to the description of a single case studies, I think this paper is of intereste for the natural hazard community and for the interaction between the communities working on landslide processes and the one interested in disaster relief. It is well written and contains clear figures.

Generally, this discussion is important because many errors and issues in scientific report on EQIL may often relate to some confusion or mixture about the aim and the use of an inventory: that is the attempt to produce an inventory somewhat for science and somewhat in an emegrency phase, whether for humanitarian response of for other rapid evaluation. As pointed out by the authors the methods can be quite opposite between an inventory to be used by reasearchers or by emergency relief practicioners. Therefore clarifying the aim and applicability of inventories that are published rapidly after a disaster would indeed be valuable and could be informed by this paper.

My comments below mainly concern some clarifications or some improvement on introduction and discussion, and should be easily addressed by the authors.

**Major comments**

1/ Could more discussion be added on how specific to Gorkha / Nepal some arguments are ? For example it is clear that the combination of rupture size (and thus affected area) and the occurence just before the monsoon, but when convection and cloud presence was already ramping up, makes the mapping and the selection of images difficult.

Earthquake in drier regions may be less affected. Smaller rupture may produce most landslides in an area covered by a few images footprints.

2/ I think the automatic landslide mapping is dismissed too rapidly in the main text, and for unconvincing reasons : "The variety and complexity of terrain and imagery makes both manual and automated mapping difficult". I appreciate that more details and discussion exist in the Table 2. But still, I think some discussions should be added on this topic.

With 5 days revisit and 10m multispectral imagery Sentinel 2 offers good prospect for medium resolution, large swath automatic detection of affected areas. Of course issues remain like the differentiation of channel bank erosion or sedimentation and landslide, or the proper detection of reactivation. Also many automatic detection methods are based on a training set and thus will need some manual mapping.

3/ As underlined by the authors, clouds are one of the most limiting factor for landslide identification. In this sense some comments on the potential of landslide detection from SAR data is missing. The recent work of on detecting landslides with Mondini 2017, suggests that such methods is within reach, even if it may mostly locate intermediate to large landslides.

**Minor comments**

P2 L 26-30 : " In the longer term, these initiatives can result in multiple inventories for the same event, further adding to the confusion. For example, Xu et al. (2014) described 14 separate landslide inventories compiled after the 2008 Wenchuan earthquake in China.After the 2015 Nepal earthquakes, there was a five-fold increase in landslide numbers between the inventories reported by Kargel et al. (2016; 4 312), Martha et al. (2016; 15 551), Roback et al. (2017; 24 915), and Tiwari et al. (2017; 14 670). "

>> This presentation is somewhat misleading : In both cases some inventories were produced and published rapidly (partly as an emergency mapping effort, for humanitarian or scientific rapid evaluation) and not necessarily in the aim of being complete. Then resolution and coverage of images mostly explain these differences. So even if more standar techniques or mapping methods are propose they may be different team publishing different partial catalogues at different time. The author could insist here or elsewhere in the text, that better flagging of the aim and validity of an inventory should be considered good practice. Additionally an evaluation of how adapted or not to humanitarian need an inventory is would certainly be positive.

P4 L 12 : focus on populous area, where landslides are " more likely to directly impact". True, but is there a discussion on the indirect impact of upstream landslides, that can also be significant, by propagating along the river system (dams, debris flow, floods).

P6 L 25- L30 : What is affected area ? The region within which landslides occured ? This one remain unknown until mapping is complete. Just above you mention the region with shaking above a limit ? On this recent work by Marc et al., 2017 propose a simple equation predicting the area of occurence of earthquake induced landslide, based on hypocenter depth, magnitude and fault location. In any case, why is the area changing from 35000 to 7500 km2 ?

P7 Line 3-4 : I understand what you mean but I think the syntax is awkward. Consider rewriting with something like: "For example, with WV2 and WV3, [...], the ability to distinguish [...] was reduced due to the lack of multispectral imagery."

P8 L 9 : I think you mean Marc et al., 2016 (seismological expression for EQIL) rather than Marc et al. 2015.

P8 L 15 : Maybe some of this discrepancies should be recall here, as well as there implications. In which case, and to which level of details should such rapid empirical model be trusted when deciding where to acquire satellite images or where to focus mapping effort ?

P8 L 27 : I don't think "verifying " is appropriate, given the example refer to a single, distant area, for model with complex distribution of landslide probabilities. Maybe "supporting" .

P10 L10 : Using difference in number of landslides is uninformattive given landslide numbers varies across several orders of magnitude. 19 000 sounds big, but for the Wenchuan earthquake it would be reasonable (only 10% of ~200 000). I think saying that you underestimated the actual numbers by a factor of ~5 is more reasonable.

Then the obvious question is, whether this was mainly due to resolution (i.e., underestimation of landslide density) or coverage (i.e., correct density but within too few mapped areas). You somewhat suggest in the next sentence that coverage and relative density is ok but absolute intensity is not.

P10 L 27 : I think that this sentence is incomplete: "[...], particularly if this area is otherwise." ?

P11 L 1 : Word missing. "Seeding an empirical landslide [model ?] with the initial rapid mapping...."

P 11 L 4 : I think it should be recall that such extrapolation based on a mapping sample, is likely to succeed only if the 100 landslides are distributed across the rupture area (not in a single catchment), and maybe across diverse setting : diverse lithologies may matter for example ...

P12 L 17-35 : Summary of automatic mapping. Ok. But constellation with high revisit frequency and medium resolution (e.g. Sentinel 2, 5 days, 10m, or Rapid Eye, ~5 days, 5m) do offer a good balance between swath size, revisit and detection capabilities. This is useful for manual mapping, but also potentially for automated detection and delineation (e.g., Stumpf, 2017), even if it may need some manual correction and edition of the detected polygons. Indeed, to me this method provide a pre classified image, allowing faster mapping, but needing significant edition.

I think such aspects may be better inroduced in the introduction and in the discussion they are also relevant to your section 4.4, because some automated classification of the image may well shorten the analysis time when cloud free images are available.

P13 L 10-15 : Sentinel 2 A and B are now operating. So I think these sentence could be shorten.

TABLE 2 : Keefer 1984 i snot in reference list. Marc et al., 2016 and Marc et al., 2017, provides considerably improved estimates of total landsliding and landslide distribution area with initial earthquake parameters and preferably some assumptions on fault rupture location (informed by moment mechanism and scalings available in the literature)

Discussion of the potential of SAR analyses exist in this table but not in the main text. A short discussion in the main text, including comments about Mondini 2017, is missing. see major comments.

The term "full inventory" is ambiguous as it described all attempt at systematically mapping landslides. But Kargel 2016, This Study, and Roback 2017 / Martha 2017, do not only differ by the way landslide are mapped (points, polylines, polgons, respectively) but also by their completeness, with the latter two having 5 times more slides than the former two.

**FIGURES**

Fig 1 : Decision tree for image selection... Could you mention quantitative criterion ? Angle of Nadir ? Resolution ? 0.5m ? 5 m ? 15m ? I know such values are not necessary strict but some indications could be given.

Figure 2: Interesting.

Figure 3-4-5-6 : I think it may be very interesting and visually effective to show in this figure the

footprints of images progressively used for mapping, maybe color coded by their resolution. See for example Fig 2 B in Xu et al 2014. It would allow to visualize the evolution of the area mapped at various resolution.

REFERENCES

Budimir 2014 : Something strange in author list. 3 authors but no initials for the last 2?

Kirschbaum 2010 : Given that you cite this work referring to the landslide database, I think you should cite the following references, and not the one you do (that is an analysis of the global dataset).

Kirschbaum, D. B., Adler, R., Hong, Y., Hill, S. and Lerner-Lam, A.: A global landslide catalog for hazard applications: method, results, and limitations, Nat Hazards, 52(3), 561–575, doi:10.1007/s11069-009-9401-4, 2009.

Parker 2017 : Is in NHESS Discussion, so far, although it may complete publication before this study.

Other references used in this review :

Marc, O., Meunier, P. and Hovius, N.: Prediction of the area affected by earthquake-induced landsliding based on seismological parameters, Nat. Hazards Earth Syst. Sci., 17(7), 1159–1175, doi:10.5194/nhess-17-1159-2017, 2017.

Mondini, A. C.: Measures of Spatial Autocorrelation Changes in Multitemporal SAR Images for Event Landslides Detection, Remote Sensing, 9(6), 554, doi:10.3390/rs9060554, 2017.

André Stumpf, O. Marc, J-P. Malet and D. Michéa, Sentinel-2 for rapid operational landslide inventory mapping, EGU2017-4449

Xu, C., Xu, X., Yao, X. and Dai, F.: Three (nearly) complete inventories of landslides triggered by the May 12, 2008 Wenchuan Mw 7.9 earthquake of China and their spatial distribution statistical analysis, Landslides, 11(3), 441–461, doi:10.1007/s10346-013-0404-6, 2014.

---

## Author Comment (AC1) · 6 Sep 2017

We are grateful to the reviewer for the highly constructive comments provided and the interest shown in the distinction between landslide mapping for disaster response and landslide mapping for scientific purposes. Here, we provide a response to both the major and minor comments in an effort to clarify the amendments that we intend to make, which are italicised throughout.

**Major comments**

Section 4.1: The reviewer highlights that the scale of the landslide affected area and the presence of considerable amounts of cloud cover are key drivers of the approach taken to map landslides. The approach may, therefore, be less suited to mapping in drier settings or over smaller areas. We intend to add this to our discussion, highlighting that the offset in timing between modelling and empirical mapping is likely to reduce in such instances. Further, the decision tree created for image selection (Fig. 1) can be amended to ignore reference to cloud cover. However, we feel that the general chronology of landslide assessment outputs, combined with the relative importance of image characteristics remains unchanged.

*... This is an attempt to identify and develop common standards for rapid SEM for landslide-triggering events that can effectively inform humanitarian response. Prior to this, it is important to consider the wider application of the SEM approach described above. This approach was heavily determined by the scale of the rupture and the presence of cloud cover in the run up to the South Asian monsoon, both of which necessitated the collection of a considerable number of images and a means of prioritising them. In drier regions or following earthquakes or rainfall that affects a much smaller area, the chronological order of outputs is unlikely to change; however, the offset in timing between initial landslide models and the mapping of landslides using either radar or optical satellite imagery is likely to increase. The 2016 Kaikoura earthquake, New Zealand, ruptured an area $200 \times 60$ km in size, similar to the $120 \times 80$ km rupture during the Gorkha earthquake. Due to cloud-free conditions and the availability of short return interval Sentinel-2 imagery, a preliminary landslide map of 1 092 landslides was released three and a half days after the earthquake with a subsequent map of 5 875 landslides within two weeks (Sortiris et al., 2016). A smaller affected area and absence of cloud cover also requires amendment to the image selection decisions in Fig. 1, such that image cloud cover and look angle are considered less important. However, the availability of imagery in Google Earth remains critical, and the order of importance of the spectral resolution, spatial resolution, and swath widths are likely to remain unchanged. In arid environments, the occurrence of landslides may be less detectable by spectral changes to the land surface than by morphological changes. A judgement may, therefore, be required as to the relative importance of image spectral and spatial resolution.*

Section 4.3: The reviewer highlights that the development of automated and semi-automated mapping techniques, combined with the availability new satellites, such as Sentinel-2, requires greater attention. Though Sentinel-2 imagery was not available following the Gorkha earthquake, having used this for subsequent landslide mapping, we agree that reference to the potential for short return interval, medium spatial-resolution, high spectral resolution imagery requires discussion and intend to add this. We feel that automated mapping is ideally placed to sit (chronologically) between landslide modelling and empirical landslide mapping. However, for the purpose of disaster response, its reliability varies considerably (for example based on the time of year) and requires considerable manual validation prior to dissemination (as presented in the paper).

*... Automated and semi-automated methods hold potential for more time efficient landslide mapping, with the speed gain over manual methods increasing with the area to be mapped. In the context of landslide-triggering disasters, these methods can be broadly divided into those that detect landslides using post-event images, and those that rely on change between pre- and post-event images. Analysis of post-event imagery, often through supervised and unsupervised classification, draws upon the spectral signature of each pixel to distinguish failures from other land cover types. Techniques that define landslides using pre- and post-disaster imagery rely on changes to the spectral signature of each pixel, most often quantified as the Normalised Difference Vegetation Index (NDVI) or the image spectral angle. However, a reliance on the spectral response of a landscape to mass-wasting has two disadvantages for landslide mapping. First, discernible changes in the spectral response may only occur as a result of high-velocity failures induced by a single event, or as a result of failures occurring within densely vegetated areas. Second, misclassification of channel bank erosion and*

*fluvial sedimentation, instead of landsliding, as well as the misidentification of reactivations may also occur. Ultimately, these techniques typically generate many more landslides than manual image interpretation due to the misclassification of non-landslide land cover types and the division of large landslides into multiple fractions, owing to the pixel-based approach to image segmentation (Borghuis et al., 2007). While the increasing availability of VHR imagery directly enhances the accuracy of manual landslide mapping, the results of automated and semi-automated pixel-based methods that have used VHR imagery are susceptible to large spectral variance between pixels, creating intra-class variabilities, and are more sensitive to coregistration errors (Moine et al., 2009; Martha et al., 2010; Mondini et al., 2011). The development of object-based image analysis (OBIA) approaches overcomes many of the issues associated with pixel-based classification, by accounting for other metrics such as color, texture, shape and topography (Stumpf and Kerle, 2011). Such techniques may benefit considerably from the rich spectral information gathered by medium resolution sensors, such as Sentinel-2, and short revisit periods that enable access to pre-event datasets. However, the selection of useful object metrics varies from case to case and the time required to prepare the algorithms, refine the parameters, and manually validate the results may exceed that required for manual mapping of the entire area. While it can be argued that the benefit of automated methods over manual methods increases with the area to be mapped, larger areas increase the reliance upon imagery from a variety of sensors. The application of semi-automated and automated mapping with variable image characteristics and quality is yet to be reported. Future research into the use of Sentinel-2 imagery in (semi-)automated landslide mapping is therefore required, but may yield an important medium between manual landslide mapping and landslide models in the aftermath of a trigger event (e.g. Stumpf et al., 2017).*

**Section 4.3:** It is noted that SAR is not included in the paper. The authors agree that this requires discussion and will, therefore, be examined in relation to mapping the potential of large-scale failures to result in potentially hazardous damming or aggradation of the channel network.

*… In instances where cloud cover is prominent, the use of satellite-borne radar also has the potential to provide an assessment of large landslides that sits between landslide models and manual mapping from optical imagery. Large failures may be rapidly identified by large morphological changes, such as shifts in the channel network. Alternatively, a large-scale shift in the dielectric constant of the slope, as vegetation is removed, may be detected by changes to the amplitude of the backscattered waves (Jin et al., 2009; Mondini et al., 2017). In this manner, SAR amplitude/intensity images have been used to map single landslides at the slope scale (Raspini et al., 2015; Plank et al., 2016) and, more recently, at the catchment scale following triggering events (Casagli et al., 2016; Mondini et al., 2017). However, SAR imagery requires a considerable amount of complex pre-processing and the accuracy of change is affected by the image acquisition geometry, which can be sub-optimal in mountainous regions.*

**Minor comments**

P2 L26-30: The reviewer identifies the statement that, in the longer-term, uncoordinated mapping efforts result in multiple different inventories of the same event as misleading. In the case of the Gorkha earthquake, we highlight that a five-fold increase in landslide numbers occurred within published inventories. While we agree that some of these inventories were created in the aftermath of the disaster, the scientific inferences made from them assume that their coverage of the affected area is complete. We agree that the flagging of the inventory aim and resolution of mapping should be advocated in the text.

*… While some of these inventories were created in the immediate aftermath of the disaster, their use for scientific purposes nevertheless assumes complete coverage of the affected area. The resolution of mapping and the approach taken should therefore be stated clearly alongside the purpose of the inventory.*

P4 L12: Our initial effort focussed on the relatively populous middle-Himalaya. As noted by the reviewer, debris flows, flooding and landslide dams that are sourced in the Higher Himalaya also have potential hazardous impacts in this region.

Given that our mapping included a number of debris flows, the length scale of their impacts were covered in the mapping. We acknowledge the importance of landslide dams in P9L5, but intend to clarify that, once identified, we monitored dams in subsequent imagery until they had breached. With regard to flooding, we intend to add this into the text. While our expertise does not lie in hydrology, and the, the landslide data was made available for others to use to assess the potential for flooding.

*… Given the potential for secondary earthquake hazards with downstream impacts, the high Himalaya were not omitted from the mapping. Our mapping coverage was consistent with the length scale of debris flows runouts, and initial searches for landslide dams were paramount. Once identified, these were monitored until breached.*

P6 L25-30: The reviewer seeks clarification over the phrasing '… small in comparison to the area that experienced shaking sufficient to trigger landsliding (~35 000 km$^2$)'. As noted by Marc et al. (2017), this definition is often variable within published coseismic landslide studies. The estimate was initially made in the aftermath of the event, referring to the mountainous area of Nepal that experienced moderate or stronger shaking according the USGS ShakeMap output, as well as from initial modelled probabilities > 0.5 (see: http://ewf.nerc.ac.uk/2015/04/25/nepal-earthquake-likely-areas-of-landsliding/). This matches with an envelope containing all mapped landslides at the end of mapping, irepective of landslide density within the envelope. As noted by Marc et al. (2017), this definition of the landslide-affected area is similar to that by Keefer (1984) and Hancox et al. (1997). We acknowledge that a more clear definition is required and have amended the following statements in light of this.

*P6 L22 … Given the prevalence of cloud cover and off-nadir viewing angles, imagery was drawn upon from a wide range of sensors. Based upon the mountainous areas of Nepal that experienced moderate to severe shaking, as estimated by ShakeMap, the area of shaking sufficient to trigger landslides was approximated at 35 000 km$^2$. This estimate was supplemented by the spatial distribution of modelled landslide probabilities > 0.5 (see: http://ewf.nerc.ac.uk/2015/04/25/nepal-earthquake-likely-areas-of-landsliding/). With the exception of the EO-1 Advanced Land Imager (ALI) and Landsat 8, the swath width of sensors such as WorldView-2 (16.4 km at nadir) and WorldView-3 (13.1 km at nadir) were small in comparison to this area, and so large numbers of relatively small-footprint images were needed for complete coverage.*

*P6 L28 … While having a high spatial resolution (~3 m) and short return period, PlanetLabs imagery had a small image footprint (~50 km$^2$) relative to the affected area.*

Our subsequent description of the polyline used to delimit the southernmost extent of landsliding on P9 L15-18 refers to the zone of most intense landsliding. The area above this line matches the final zone of intense landsliding mapped by ourselves and subsequently by Martha et al (2016), ca. 12 000 km$^2$. This has been amended and added to the text in the following locations.

*P9 L15 … The aim of this was to delineate the southernmost limit of major landslide disruption, and hence the likely northern limit of unimpeded road access, using the predominantly north – south oriented drainage network. This was mapped as a solid line where the limit was observed and a dashed line where the limit was inferred in the absence of imagery. Subsequent mapping showed this line to be an accurate estimate, with the area of intense landsliding (ca. 12 000 km$^2$) matching our own final product and that of Martha et al. (2016).*

*P4 L19-21… A rapid appraisal of the first available imagery suggested that the most intense landsliding occurred in an E-W swath located north of the Kathmandu Valley, covering a large proportion of Western and Central Nepal (12 000 km$^2$).*

P7 L3-4: The reviewer notes that the syntax of this line is awkward, with the following suggestion: 'For example, with WV2 and WV3, [...], the ability to distinguish [...] was reduced due to the lack of multispectral imagery.' In light of this, we have made the amendment presented below.

*... In the case of WorldView-2 and WorldView-3, although panchromatic imagery provides greater spatial resolution, the ability to distinguish vegetation from freshly exposed bedrock and regolith in landslide scars was reduced due to the lack of multispectral imagery.*

P8 L9: The reviewer highlights a mistake in the citation of Marc *et al*., which is corrected in the below statement.

*... and local site geology (Meunier et al., 2008; Parker et al., 2015; Marc et al., 2016).*

P8 L15: The reviewer asks for more information regarding the discrepancies between landslide models and the landslide mapping. We feel that these discrepancies are beyond the scope of this paper given that such an analysis would require comparisons of multiple different landslide models and that these have been discussed by others, such as Gallen et al. (2016). However, we intend to add that the overestimation of landslide probabilities south of Kathmandu in Sivalik Hills occurred.

*... Comparisons between predicted landslide density and observed landslide density have since highlighted some important discrepancies (Gallen et al., 2016), including an overestimation of landsliding to the south of Kathmandu in the Sivalik Hills.*

P8 L27: We agree that '… verifying the extent of landsliding predicted by the landslide models by examining small gaps in cloud cover within satellite imagery', requires clarification. As noted by the reviewer, verifying such a model would not be possible if the area viewed refers to a single, distant area with a complex distribution of landslide probabilities.

*... This gap in cloud was ~120 km from the epicentre and provided an initial assessment of the nature, type and density of landsliding in the area, as well as supporting modelled estimates of the extent of the area affected by landsliding.*

P10 L10: As noted by the reviewer, quantifying our underestimation of landsliding as the difference in number is uninformative. A ratio or percentage is suggested and amended accordingly below.

*... Underestimated the number of landslides by a factor of five (up to ~19 000 landslides). However, the spatial pattern and relative intensity closely adheres to those described in both Martha et al. (2016) and Roback et al. (2017), suggesting that small areas of cloud cover, the spatial resolution of mapping, and the distinction of separate failures was lower in our approach.*

P10 L27: The reviewer highlights a missing word at the end of the sentence, which is added below.

*... particularly if this area is otherwise inaccessible.*

P11 L1: The reviewer suggests adding the missing word 'model'. This is added below

*… Seeding an empirical landslide model with the initial rapid mapping*

P11 L4: The reviewer highlights that training a landslide model using a small number (~$10^2$) of landslides requires that they are well distributed across the affected area and across diverse lithological settings. This is discussed by Robinson et al. (2017), to which reference is added below.

*… Robinson et al. (2017) found that a small number of landslides could be used to train landslide models as long as their spatial distribution covered a large portion of the affected area.*

P12 L17-35: As with major comment #2, the reviewer highlights that the discussion of automated mapping requires broadening, for example, with the potential of Seninel-2 imagery. While we do make reference to the potential of Sentinel-2 imagery at P13 L7, we have added this to the discussion of (semi-)automated methods in the amendment of major comment #2.

P13 L10-15: Given that both Sentinel-2a and 2b are now operating, the reviewer notes that our description of the reduction in revisit time from 10 days to five can be shortened. We have revised this sentence in light of this.

*… In addition, the shorter return period (five days for Sentinel-2a and -2b, compared to 16 days for Landsat 8) will increase the probability of observing the ground through gaps in cloud cover, reducing the time needed to process outputs. Our effort demonstrated that once imagery is available, mapping can be rapid (two to three days), given suitable capacity.*

**Amendments to Table 2**

The reviewer comments that the phrase 'full inventory' is ambiguous. We have revised instances where this is used below.

P23 *… Can be assessed qualitatively without the need for full coverage with each individual landslide identified*

P23 *… Potentially rapid generation of a polygon-based landslide inventory across the entire affected area*

P24 *… Not reliant on having a landslide inventory of full coverage*

P24 *… relatively quick to create an inventory of full coverage*

P24 *… Landslide mapping: Full coverage*

**Amendments to Figures**

Fig. 1: The reviewer suggests that indicative values should be added to the decision tree for image prioritisation. While certain characteristics were quantified during our selection process (e.g. cloud cover < 20%), we feel that the importance (and potential for use) of this figures lies in the order in which image characteristics are prioritised for this particular type of SEM. We are therefore hesitant to add indicative values, given that such values may not be applicable to earthquake-induced landsliding in other settings.

Fig. 3-6: We agree that showing our image footprints through time could be useful; however, the mapping was not linear through time in terms of downloading imagery from HDDS Explorer and mapping from it in ArcMap. This was because bundles of mosaicked imagery were also iteratively released through Google Crisis. Furthermore, while the spatial resolution and timing of imagery is likely to broadly correlate with mapping progress, we feel that the considerable variability in image distortion and cloud cover would perhaps be overlooked.

**Amendments to References**

Budimir, M.E.A., Atkinson, P.M., and Lewis, H.G.: Earthquake-and-landslide events are associated with more fatalities than earthquakes alone, Nat. Haz., 72(2), 895–914, doi:10.1007/s11069-014-1044-4, 2014.

Keefer, D.K.: Landslides caused by earthquakes. Geol. Soc. Am. Bull., 95(4), 406-421, doi: 10.1130/0016-7606(1984)95<406:LCBE>2.0.CO;2, 1984.

Kirschbaum, D.B., Adler, R., Hong, Y., Hill, S. and Lerner-Lam, A.: A global landslide catalog for hazard applications: method, results, and limitations, Nat Hazards, 52(3), 561–575, doi:10.1007/s11069-009-9401-4, 2009.

Marc, O., Meunier, P. and Hovius, N.: Prediction of the area affected by earthquake-induced landsliding based on seismological parameters, Nat. Hazards Earth Syst. Sci., 17(7), 1159–1175, doi:10.5194/nhess-17-1159-2017, 2017.

Mondini, A.C.: Measures of Spatial Autocorrelation Changes in Multitemporal SAR images for event landslides detection, Remote Sensing, 9(6), 554, doi:10.3390/rs9060554, 2017.

Stumpf, A., Marc, O., Malet, J.P., and Michea, D.: Sentinel-2 for rapid operational landslide inventory mapping, EGU General Assembly Conference, 23-28 April, 19, 4449, 2017.

---

## Referee Comment (RC2) · Anonymous Referee #2 · 19 Sep 2017

The paper presents and discusses an interest topic, i.e., the emergency mapping of landslides, with examples from the 2015 Nepal earthquake. I have some concern about the typology of publication. Indeed, the paper does not presents a typical research based on the analysis of data, but rather a speculation on the problems related to landslide mapping in emergency condition. For this reason, I believe that the paper could be accepted for publication as a brief communication (after significant shortening) and not as a research paper. I'll try to substantiate my opinion hereafter. 1 -The first and main problem is that the paper is based on a single case study. The four figures presents four successive steps of advancements of the inventory, and have been published online first, and they are probably still available in the HDX site (https://data.humdata.org/). The real problem of the paper is that the general conclusions on the emergency map-

ping of landslides are rooted in this specific case study. While they pretend to be "general", they are indeed "specific". A few examples. One of the conclusion is that manual mapping is not fast enough (session 4.2), and a faster approach is needed (Robinson et al, 2017 is cited as an example). Indeed, the delay in mapping in Nepal was mainly due to the clouds that covered the sky soon after the earthquake. This required a few days to be have good images available. However, this is not always the case. For instance, if good images were available since the first day, one could have mapped hundreds of landslides within 4 or 5 days (consider that a good geomorphologist could map tens of landslides a day). Hence, manual mapping is not the issue. The issue is how good the weather is (for instance) and how lucky we are in having satellites ready to take the images on time. Another conclusion (session 4.3) is that linear mapping was a good compromise between velocity and the need to assess the landslides size, even roughly. Part of reason for this choice is that the georeferencing of Google Crisis maps was very poor, hence hampering a meaningful mapping of polygons. Again, this is not always the case. In the future, we could expect Google to provide better and better image datasets, and we could expect to have a good georeferencing soon after the event. Hence, the conclusion is true for this case study TODAY, but it is not general and probably not completely true for the future.

2 - A second significant problem is that part of the speculations are not supported by any analysis. For example, the potential of crowd-sourced information. I agree that this may be relevant in the future, but the case study does not say anything about that. The same for other speculations. For instance, the accompanying information of the output (last 4 numbered points in the discussion) are not discussed based on the present case study. A third example regards the comparison of the inventories (session 4.1). The authors state that, even if their inventory has less landslides than the others, it still holds value as a rapid assessment etc. etc. Again, this could be true but it is speculated without any analysis of the data. In such case, data analysis could have been done by trying to overlap polygons to identify positional mismatch and overlapping ratio.

3 – the third problem is that large part of the paper is not transferable to other similar emergency situations. For instance, the data selection (session 2.2), and the mapping platform (session 2.3) are very site specific and may be different for other case studies. Hence, a long description of these issues are not relevant, and may be strongly reduced.

My conclusion is that the authors should resubmit the paper as a brief communication after significant shortening. They could keep session 1.1 and 1.2. The should strongly reduce chapter 3 (maybe it can be moved in the supplementary materials) and chapter 4 (just the figures with appropriate long captions could be enough). Finally, they could save the discussion, stressing out what is of general purposes and what is case-study specific, and trying to figure out what could be the issue in the near future.
* * *

---

## Author Comment (AC2) · 2 Nov 2017

We thank the referee for providing a review of our manuscript. We note that the tone of the review perhaps reflects a misunderstanding of our primary aim: to focus upon the process, timing and decisions made in generating a rapid landslide assessment immediately after an earthquake. Reviews of this type of experience, and the lessons gained from it, have not been published to our knowledge. Based upon our own effort following the 2015 Gorkha earthquake, we argue that such a discussion is of value for those engaged in similar activities. Importantly, our research shows that the availability of imagery is only one of many constraints in providing a timely and useful assessment of landsliding that is of value to disaster response practitioners on the ground. Moreover, while protocols for the use of earth observation data for other geohazards are well-established, those for rapidly assessing landslides remain absent. Importantly, our inventory, and the submitted paper that describes its creation, are not intended to compare to other earthquake triggered landslide datasets. To make the unique focus of our manuscript clear, we suggest amending our title to: '*Satellite-based emergency landslide mapping: Experience and reflections from the 2015 Nepal earthquake*'.

Below we provide responses to each of the comments made by the reviewer.

**(1) The paper does not presents a typical research based on the analysis of data, but rather a speculation on the problems related to landslide mapping in emergency condition.**

We define our aim on P1L19: '*we share the lessons learned from the Gorkha earthquake, with the aim of informing the approach taken by scientists to understand the evolving landslide hazard in future events and the expectations of the humanitarian community involved in disaster response*'.

The paper reports on our experiences. While this may not constitute a typical presentation of data, in our view it is not at odds with the interests of the journal readership. Where limitations in the method are identified, they are based on our own experiences rather than speculative. The prioritisation of image attributes to enhance the efficiency of manual mapping (Fig. 1), and the chronology of landslide assessment outputs (Table 2) are two of a number of instances where we feel our experiences are of value to those responding to future landslide disasters. Our discussion is intended to initiate further discussion around the timeliness of landslide mapping relative to the rapid rollout of a typical disaster response (Section 4.2), and the inevitable time-limited choices that have to be made (Section 4.3). The authors have been involved in generating landslide inventories for numerous previous earthquakes for scientific ends. However, in attempting to supplement the humanitarian response following the 2015 Nepal earthquakes, we found no guidance, documented experience or reflections upon the best practice for rapid landslide assessment within the timescales described here.

**(2) The first and main problem is that the paper is based on a single case study.**

It is true that our paper builds upon a single case study but we do not believe that it constitutes a problem. A discussion of what is feasible and useful to generate in the aftermath of a disaster is important to initiate, in order to provide a more efficient, coherent, and useful response. We present general and transferable observations that we anticipate will be more widely relevant beyond a single case study.

**(3) The four figures presents four successive steps of advancements of the inventory, and have been published online first, and they are probably still available in the HDX site (https://data.humdata.org/).**

The reviewer refers here to Fig. 2 – Fig. 6. The data displayed are avaialble on HDX as polylines, and was previously disseminated to inform the post-disaster response in 2015. The maps presented in the manuscript based upon this data are entirely new, and have been formatted to fully represent the development of the mapping (e.g. density maps showing landslides per square kilometre are represented using a consistent colour ramp).

**(4) The real problem of the paper is that the general conclusions on the emergency mapping of landslides are rooted in this specific case study. While they pretend to be "general", they are indeed "specific".**

We disagree with this statement for the reasons outlined above. To reiterate, we strike a balance between learning from a detailed case study, and drawing general and transferrable conclusions. We draw the reader's attention to the findings described in Table 2, which have no specific reference to Nepal. With the possible exception of cloud cover (see '5' below), the decisions associated with image selection in Figure 1 are also not specific to this case study, and will be the same for any situation requiring rapid landslide mapping, either now or into the foreseeable future.

**(5) One of the conclusion is that manual mapping is not fast enough (session 4.2), and a faster approach is needed (Robinson et al, 2017 is cited as an example). Indeed, the delay in mapping in Nepal was mainly due to the clouds that covered the sky soon after the earthquake. This required a few days to be have good images available. However, this is not always the case. For instance, if good images were available since the first day, one could have mapped hundreds of landslides within 4 or 5 days (consider that a good geomorphologist could map tens of landslides a day).**

We refer the reviewer to Table 2, where a detailed chronology of the landslide assessments generated is described. Following the first cloud-free imagery, the production of an initial landslide assessment and inventory was available within approximately five days, reflecting broadly the timescales outlined in the reviewer's comment above. Our paper highlights that, while this is possible, a set of decisions is still necessary in terms of defining image selection, mapping protocol and outputs. Some of the most useful outputs from this assessment were derived in the first days after cloud-free imagery, which were provided without mapping individual landslides (landsliding extent, southernmost landsliding limit, hotspots of landslide impacts). We therefore argue that setting out to generate a detailed landslide-by-landslide inventory may not always be the most beneficial activity to those responding on the ground. Given the importance attached to cloud-cover in post-disaster landslide mapping, we intend to clarify its impact on the content of table 2 as follows:

P15L5: *The various means of landslide assessment that have been discussed above are summarised in Table 2. This provides a chronology of outputs that clarifies what we have found possible to achieve within the timeframes of the UN Situation Analysis and MIRA report. The timescales of what is possible will vary between events, predominantly as a function of cloud cover for landslide mapping, but the suggested timescales in Table 2 are broadly independent of this. For example, following the first cloud-free imagery after the Gorkha earthquake, the production of an initial landslide assessment and inventory was available within approximately five days, as reflected in the description of a full point inventory.*

Where an earthquake triggers thousands of landslides, we emphasise that manual mapping alone cannot respond quickly enough to meet the needs of the initial disaster responders, whether clouds are present or not. We do not advocate for landslide modelling instead of manual mapping. We highlight that initial mapping can help to refine landslide modelling (hence the reference to Robinson et al., 2017) while also providing valuable information on the scale, extent, and distribution of landslide impacts across the entire affected area within the first 72 hours (even using only small gaps in cloud). Uniquely, our discussion also draws attention to the needs of disaster response, and how they shift quickly from a broad overview to increasingly local and specific details of individual failures. A method to map thousands of individual failures as efficiently as possible has therefore been described. We also highlight the importance of manual mapping in the identification of secondary hazards, such as landslide dams. We agree that every disaster will be different and so some of the issues raised will inevitably not always play a role. However, it is important to discuss the usefulness and feasibility of various post-earthquake landslide assessments with regard to the timescales, priorities and expectations involved.

**(6) Hence, manual mapping is not the issue. The issue is how good the weather is (for instance) …**

We refer the reader to our previous responses and have clarified the impact of weather on the paper's wider applicability in response to the review by Odin Marc. Given that our review focusses on SEM for landslide assessment, it is important to note that the settings that are relevant here are steep and mountainous, increasing the potential for cloud cover. As noted on P4 L26, conditions that are ideal for landslide mapping are often not present or coincident in mountainous regions. This is particularly so if widespread landsliding has

been triggered by a storm event, to which the manual mapping method described is also applicable. In light of this, we intend to clarify the importance of cloud cover as follows:

P5L10: *Given that landslides typically occur in steep and mountainous regions, often following prolonged rainfall, the potential for cloud cover in imagery is a key consideration for associated SEM. The Nepal Himalaya, for example, are obscured by cloud between mid-June and mid-September each year, during which time an estimated 90% of annual fatal landsliding occurs (Petley et al., 2007).*

**(7) … and how lucky we are in having satellites ready to take the images on time.**

A key finding from the manuscript is that it cannot be assumed that, once an image is captured, a landslide inventory or assessment will become readily available (Section 4.4). This is a problematic assumption that raises expectations of both those producing landslide assessments, but also those who could use them. Based on the reviewer's comment, we have sought to clarify this at the end of the second paragraph of Section 4.4:

P14L17: *Our effort demonstrated that once imagery is available, mapping can be rapid (two to three days), given suitable capacity. However, we have also found that it cannot be assumed that a landslide inventory or assessment will become readily available once an image is captured. This is a problematic assumption that raises expectations of both those producing landslide assessments, but also those who could use them.*

While beyond the focus here, the return period of satellites is well known and is ever-decreasing. Sentinel-2, for example, has the potential to considerably increase the likelihood that imagery will be captured across an area within five days of a triggering event. We argue that the experiences presented and guidance for future events derived from our effort in Nepal will gain more importance in a context where satellite imagery is more readily and rapidly available. The need to make choices about the best imagery to use, and the value of different approaches to mapping remains irrespective of the frequency of image capture.

**(8) Another conclusion (session 4.3) is that linear mapping was a good compromise between velocity and the need to assess the landslides size, even roughly. Part of reason for this choice is that the georeferencing of Google Crisis maps was very poor, hence hampering a meaningful mapping of polygons. Again, this is not always the case.**

We assume the reviewer is referring to our decision to map using polylines as opposed to points or polygons. The locational accuracy of imagery for landslide mapping can be low in places, given that landslides occur in steep terrain where the angle of incidence to the sensor may be high. This has the potential to influence object characteristics (e.g. area) such that comparison between sites or cumulative statistics are of questionable value. Our decision to map polylines therefore relates to the incidence angle of high-resolution imagery in steep terrain, not to the georeferencing quality of online imagery (including Google Crisis). It is therefore a property of the situation described, which is very likely to be repeated, rather than a peculiarity of this particular event and the actors involved. We reiterate our primary reason for using polylines, which is that the approach retains information on landslide scale, location, and intersection with assets, yet is considerably more efficient to map. This is described in detail in Sections 2.3 and 4.3. In the context of disaster response, speed is of the essence.

**(9) A second significant problem is that part of the speculations are not supported by any analysis. For example, the potential of crowd-sourced information. I agree that this may be relevant in the future, but the case study does not say anything about that.**

Our focus and analysis constitutes a reflection on the mapping process that was undertaken after the Gorkha earthquake. While others have used our dataset to analyse landslide distributions (e.g. Valagussa et al., 2016), this is not our intended aim. Crowd-sourced information was a supplementary component of the Gorkha earthquake response, and is likely to increase in importance as this technology develops. While we were not able to incorporate this into our mapping, we feel it is important to identify the potential benefits and limitations.

**(10) A third example regards the comparison of the inventories (session 4.1). The authors state that, even if their inventory has less landslides than the others, it still holds value as a rapid assessment etc. etc. Again, this could be true but it is speculated without any analysis of the**

**data. In such case, data analysis could have been done by trying to overlap polygons to identify positional mismatch and overlapping ratio.**

We agree that the relative intensity and spatial distribution of our inventory could be compared to other coseismic landslide inventories collated for this event. We have since undertaken a geospatial comparison with the inventory reported by Roback et al. (2017) that allows us to qualify these uncertainties directly. While we feel that a full numerical analysis would detract from the focus of the manuscript, and is something that is well-suited for further analysis elsewhere, we will report the analysis that we have undertaken in the Supplementary Materials (see below), and report the following statements at the beginning of Section 4.1 to qualify the statements with regard to the comparison of inventories:

P11L9: *Comparing our rapidly-derived inventory with subsequent, independently collated inventories (Martha et al., 2016; Roback et al., 2017; Tiwari et al., 2017) shows that our inventory underestimated the total number of landslides by up to ~ 19 000. When compared for every 1 km$^2$ of the landslide-affected area, our inventory underestimates landslide number by an average factor of 1.8, which is broadly consistent irrespective of landslide density. However, the spatial pattern and relative intensity closely adheres to those described in both Martha et al. (2016) and Roback et al. (2017). The overall extent of the mapped landslide affected area are broadly similar (Fig. S1), covering the same geographical footprint. In addition, the locations of highest density landsliding and the southernmost limit of landsliding is consistent between the inventories.*

We again refer the reviewer to the primary purpose of our paper, to provide reflection of the generation of landslide assessment in the immediate aftermath of a large earthquake. In this context, the comparability of these datasets reinforces the value of the rapid assessment presented and discussed here.

**(11) The third problem is that large part of the paper is not transferable to other similar emergency situations. For instance, the data selection (session 2.2), and the mapping platform (session 2.3) are very site specific and may be different for other case studies. Hence, a long description of these issues are not relevant, and may be strongly reduced.**

This statement appears to summarise the reviewer's previous comments. We therefore refer the reviewer to the response above.

**References used in this response**

Petley, D., Hearn, G., Hart, A., Rosser, N., Dunning, S., Oven, K. and Mitchell, W.: Trends in landslide occurrence in Nepal, Nat. Hazards, 43, 23-44, doi: 10.1007/s11069-006-9100-3, 2007.

Roback, K., Clark, M.K., West, A.J., Zekkos, D. and Li, G.: The size, distribution, and mobility of landslides caused by the 2015 M$_w$ 7.8 Gorkha earthquake, Nepal, Geomorphology, doi: 10.1016/j.geomorph.2017.01.030, 2017.

Robinson, T., Rosser, N., Densmore, A., Williams, J., Kincey, M., Benjamin, J. and Bell, H.: Rapid post-earthquake modelling of coseismic landslide magnitude and distribution for emergency response decision support, Nat. Haz. Earth Sys., 1–29, doi: 10.5194/nhess-2017-83, 2017.

Tiwari, B, Ajmera, B and Dhital, S: Characteristics of moderate-to large-scale landslides triggered by the M$_w$ 7.8 2015 Gorkha earthquake and its aftershocks, Landslides, doi: 10.1007/s10346-016-0789-0, 2017.

Valagussa, A., Frattini, P., Crosta, G. and Valbuzzi, E.: Regional analysis of distribution of pre and post 2015 Nepal Earthquake landslides, EGU General Assembly Conference Abstracts, 17-22 April, 18, 17045, 2016.

**Supplementary materials**

Fig. S1. The number of landslides in both inventories was counted for 1 km$^2$ grid cells occupying the same spatial extents. The result is a map of landslide density, represented as the number of landslides per square kilometre. Given the fivefold increase in the number of landslides mapped by Roback et al. (2017), each grid cell was normalised by the maximum density of landsliding for that inventory (27 for the Durham inventory, 84 for the Roback inventory). This provides a comparable spatial distribution of landslide intensity. This distribution is of greater importance in the context of disaster response than the absolute number of landslides, which inevitably varies with the method of mapping and the level of detail involved.

[Figure]

---

## Author Response (AR1)

Dear Dr Malet,

Many thanks for your comments and suggestions for our manuscript. We have added to our manuscript a 'cooking recipe' for manual mapping from optical imagery post-earthquake. This is tracked and can be found in Section 4.6. Within this, we have made several references to the Copernicus Emergency Management Service products.

Please do not hesitate to contact us should you seek clarification or further amendment.

Sincerely,

Dr Jack Williams